# Quantitative proteomic analysis reveals posttranslational responses to aneuploidy in yeast

Noah Dephoure[1], Sunyoung Hwang[2], Ciara O'Sullivan[2], Stacie E Dodgson[3], Steven P Gygi[1], Angelika Amon[3], Eduardo M Torres[2]*

[1]Department of Cell Biology, Harvard Medical School, Boston, United States; [2]Program in Gene Function and Expression, University of Massachusetts Medical School, Worcester, United States; [3]David H. Koch Institute for Integrative Cancer Research, Howard Hughes Medical Institute, Massachusetts Institute of Technology, Cambridge, United States

**Abstract** Aneuploidy causes severe developmental defects and is a near universal feature of tumor cells. Despite its profound effects, the cellular processes affected by aneuploidy are not well characterized. Here, we examined the consequences of aneuploidy on the proteome of aneuploid budding yeast strains. We show that although protein levels largely scale with gene copy number, subunits of multi-protein complexes are notable exceptions. Posttranslational mechanisms attenuate their expression when their encoding genes are in excess. Our proteomic analyses further revealed a novel aneuploidy-associated protein expression signature characteristic of altered metabolism and redox homeostasis. Indeed aneuploid cells harbor increased levels of reactive oxygen species (ROS). Interestingly, increased protein turnover attenuates ROS levels and this novel aneuploidy-associated signature and improves the fitness of most aneuploid strains. Our results show that aneuploidy causes alterations in metabolism and redox homeostasis. Cells respond to these alterations through both transcriptional and posttranscriptional mechanisms.

*For correspondence: eduardo. torres@umassmed.edu

**Reviewing editor**: Ivan Dikic, Goethe University, Germany

## Introduction

Aneuploidy, a condition of having a chromosome number that is not an exact multiple of the haploid complement, has detrimental effects on the development of all eukaryotic organisms where it has been studied (*Torres et al., 2008*). In humans, aneuploidy is the major cause of spontaneous abortions and mental retardation, and it is found in most solid tumors and leukemias (*Weaver and Cleveland, 2006*; *Nagaoka et al., 2012*).

To gain insight into the consequences of aneuploidy at the cellular level and its role in tumorigenesis, we studied the effects of gaining an extra chromosome in haploid yeast cells (henceforth disomes). We showed that yeast cells harboring an extra chromosome share a number of phenotypes including impaired proliferation, increased genomic instability, traits indicative of proteotoxic stress and a gene expression signature known as the environmental stress response (ESR), which is associated with slow growth and stress (*Gasch et al., 2000*; *Torres et al., 2007*; *Sheltzer et al., 2012*). Importantly, these aneuploidy-associated stresses are also present in aneuploid mammalian cells (*Williams et al., 2008*; *Stingele et al., 2012*). Based on these findings, we proposed that the aneuploid state has general consequences beyond those conferred by the increased copy number of specific genes.

A key feature of the aneuploid condition is its impact on protein homeostasis. Aneuploid yeast cells are prone to aggregation of both endogenous proteins and ectopically expressed hard-to-fold

**eLife digest** Nearly all tumor cells contain abnormal number of chromosomes. This state is called aneuploidy, and can also cause embryos to be miscarried, or to be born with severe developmental disorders.

Proteins are produced from the genes contained within chromosomes, and so cells with too many chromosomes produce too many of some proteins. How do these cells cope with this excess? Previous work identified one strategy where a gene called *UBP6* is mutated to prevent it from working correctly. The *UBP6* gene normally encodes a protein that removes a small tag (called ubiquitin) from other proteins. This tag normally marks other proteins that should be degraded; thus, if *UBP6* is not working, more proteins are broken down.

Dephoure et al. investigated the effect of aneuploidy on the proteins produced by 12 different types of yeast cell, which each had an extra chromosome. In general, the amount of each protein produced by these yeast increased depending on the number of extra copies of the matching genes found on the extra chromosome. However, this was not the case for around 20% of the proteins, which were found in lower amounts than expected. Dephoure et al. revealed that this was not because fewer proteins were made, but because more were broken down. These proteins may be targeted for degradation because they are unstable, as many of these proteins need to bind to other proteins to keep them stable—but these stabilizing proteins are not also over-produced.

Aneuploidy in cells also has other effects, including changing the cells' metabolism so that the cells grow more slowly and do not respond as well to stress. However, Dephoure et al. found that, as well as reducing the number of proteins produced, deleting the *UBP6* gene also increased the fitness of the cells. Targeting the protein encoded by the *UBP6* gene, or others that also stop proteins being broken down, could therefore help to reduce the negative effects of aneuploidy for a cell. Whether targeting these genes or proteins could also help to treat the diseases and disorders that result from aneuploidy, such as Alzheimer's and Huntington's disease, remains to be investigated.

proteins (*Oromendia et al., 2012*). Furthermore, they exhibit increased sensitivity to inhibitors of protein translation, degradation, or folding (*Torres et al., 2007*). Aneuploid mammalian cells are also sensitive to compounds that interfere with protein quality control mechanisms such as chaperone activity or autophagy (*Tang et al., 2011*). These observations suggest that the proteomic imbalances caused by an aneuploid karyotype disrupt protein homeostasis.

How do aneuploid cancer cells overcome the detrimental effects of aneuploidy? We hypothesized that they may harbor mutations that suppress the adverse effects of aneuploidy. We showed that such mutations indeed exist. In a selection, we identified mutations that improve the fitness of aneuploid yeast strains. Among them was a loss-of-function mutation in the gene encoding the deubiquitinating enzyme Ubp6, which results in enhanced proteasomal degradation (*Hanna et al., 2006*; *Torres et al., 2010*). Deletion of *UBP6* improved the fitness of several disomic yeast strains under standard growth conditions and attenuated the proteomic changes caused by aneuploidy. Whether deletion of *UBP6* improves the fitness of aneuploid yeast strains in general or whether it is restricted to specific aneuploid karyotypes is not known, nor is the mechanism whereby deletion of the *UBP6* gene suppresses the proliferation defect associated with aneuploidy.

Here we investigate the impact of aneuploidy on the cell's proteome and how Ubp6 dampens the impact of the aneuploid condition. Our studies show that protein abundances largely scale with gene copy number but that ~20% of proteins encoded by genes present on additional chromosomes are attenuated. The majority of the attenuated proteins are components of multi-subunit complexes. This finding has implications not only for understanding how cells respond to aneuploidy, but also for how protein complexes are formed and maintained in euploid cells. Importantly, our analysis revealed the existence of both transcriptionally and post-transcriptionally mediated protein expression changes indicative of slow growth as well as oxidative and metabolic stress. Deleting *UBP6* attenuates the impact of aneuploidy on the proteome and fitness of all aneuploid yeast strains analyzed, highlighting the importance of proteasomal degradation for aneuploidy tolerance.

## Results

### Cellular protein composition is altered by aneuploidy

To understand the global consequences of aneuploidy on the proteome, we used stable isotope labeling of amino acids in cell culture (SILAC) (*Ong et al., 2002*) and liquid chromatography—tandem mass spectrometry (LC-MS/MS) to profile protein abundances in 12 different disomic strains (Disomes I, II, V, VIII, IX, X, XI, XII, XIII, XIV, XV and XVI) (*Figure 1A,B*, 'Materials and methods'). These experiments revealed quantitative information for ~70–80% of all verified open reading frames (ORFs) in the disomic strains relative to wild-type cells (*Figure 1B*, *Figure 1—source data 1*). A comparison of wild-type/wild-type cells showed a standard deviation (SD) of the $\log_2$ ratios equal to 0.35 (*Figure 1—figure supplement 1*). Analysis of the protein abundances encoded by genes on the duplicated chromosomes of all 12 disomic strains demonstrates that on average protein levels increased approximately twofold (*Figure 1B*). This correlation is apparent when $\log_2$ ratios of protein levels of disomic strains relative to wild-type cells are sorted by the chromosomal position of their encoding genes (*Figure 1B*).

Growth conditions can significantly influence gene expression. Because aneuploidy increases genomic instability including higher rates of chromosome loss, disomic strains are grown in medium that selects for the presence of the duplicated chromosome ('Materials and methods'). In addition, SILAC relies on the use of synthetic medium supplemented with 'heavy' or 'light' amino acids. To determine whether growth conditions affect the proteome composition of disomic strains, we grew cells in rich medium (YEPD) for a small number of generations and utilized isobaric tandem mass tag (TMT)-based quantitative mass spectrometry ('Materials and methods', *Figure 1C*) to assess the proteome of the 12 disomic yeast strains. In total, we obtained quantitative information for ~65–74% of all verified ORFs in the disomes relative to wild-type-cells (*Figure 1D*, *Figure 1—source data 1*). As seen in the SILAC-based quantifications, analysis of protein levels of genes encoded on the duplicated chromosomes of 12 disomes showed an average increase of ~twofold (*Figure 1D*). The $\log_2$ ratios of control wild-type/wild-type cells showed low noise and high reproducibility in the data (SD of $\log_2$ ratios = 0.2, *Figure 1—figure supplement 1B*). Importantly, comparison of the changes in gene expression and protein abundances of disomes compared to wild-type cells grown under similar conditions revealed significant correlations between mRNA and protein levels (*Figure 1—figure supplement 2*, *Figure 2—source data 1*). These results indicate that on average, increases in gene copy number lead to proportional increases in mRNA and protein levels independent of growth conditions.

### Several proteins encoded by genes located on the duplicated chromosomes are attenuated

Dosage compensation, where a change in gene dosage does not lead to a corresponding change in protein levels, is common for genes located on sex chromosomes (*Lee and Bartolomei, 2013*). Whether dosage compensation also occurs on autosomes and if so, which genes are affected and how it is brought about are critical questions not only to understand the effects of aneuploidy but also to understand how protein homeostasis is maintained in normal cells. Our set of disomic yeast strains, which comprises duplications of 12 of the 16 chromosomes (corresponding to 73% of the yeast genome), allowed us to address this question. We grew the 12 disomic strains in rich medium, split the cultures and analyzed mRNA and protein levels. In total, we obtained quantitative information for both mRNA and protein, reported as $\log_2$ ratios, for 2,581 genes located on duplicated chromosomes (*Figure 2A,B*) and 39,011 paired measurements for genes on non-duplicated chromosomes (*Figure 2C,D*). The ratios of mRNA levels of duplicated genes fit a normal distribution with a mean increase of 1.9-fold (SD = 0.3 and $R^2$ = 0.99, *Figure 2A*). Parallel analysis of the corresponding protein changes did not fit as well to a normal distribution ($R^2$ = 0.96, Pearson's mode skewness = −0.12). Nonlinear regression analysis of the protein data best fit a sum of two normal distributions; one with a mean increase of twofold, the other with a significantly reduced mean increase of ~1.6-fold ($R^2$ = 1.00, *Figure 2B*). In contrast, analysis of both mRNA and protein changes of non-duplicated genes showed nearly perfect normal distributions ($R^2$ = 0.99, *Figure 2C,D*). These analyses indicate that although acquisition of an extra chromosome leads on average to twofold increases in mRNA levels of the duplicated genes, a large and statistically significant number of proteins do not increase proportionally with copy number. Importantly, neither the growth conditions nor the quantitative approach affected the degree of attenuation, as analysis of mRNA and protein levels from cells grown in selective medium and analyzed by SILAC showed similar results (*Figure 2—figure supplement 1*).

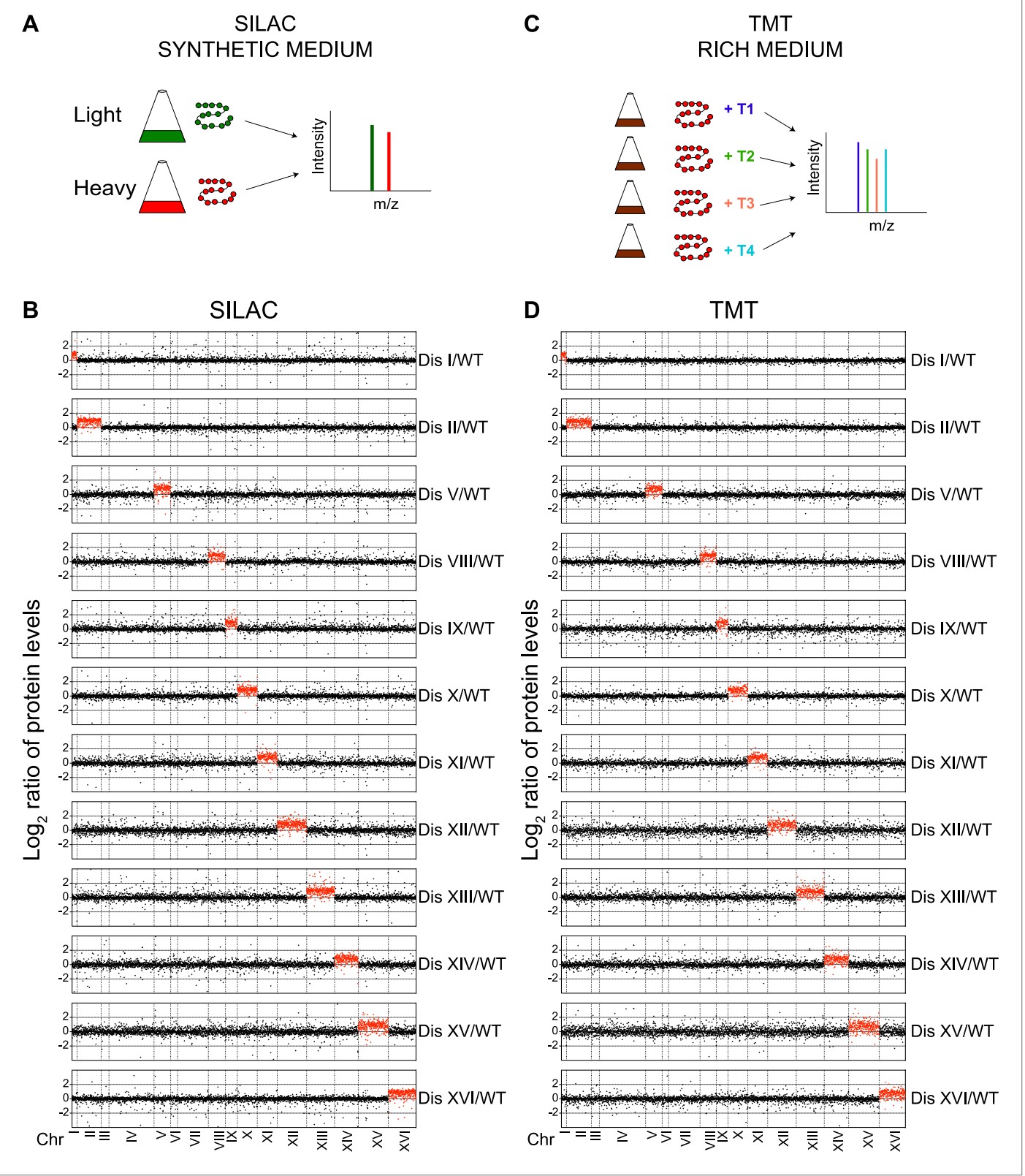

**Figure 1**. Proteome quantification of aneuploid yeast strains. (**A**) Schematic of the approach utilized in stable isotope labeling of amino acids in cell culture (SILAC) and liquid chromatography—mass spectrometry. (**B**) The plots show the log$_2$ ratio of the relative protein abundance of disomes compared to wild-type cells grown in synthetic medium. Protein levels are shown in the order of the chromosomal location of their encoding genes. Protein

*Figure 1. Continued on next page*

*Figure 1. Continued*

levels of duplicated chromosomes are shown in red. (**C**) Schematic of the approach utilized in isobaric tandem mass tag (TMT)-based quantitative mass spectrometry. (**D**) The plots show the $\log_2$ ratio of the relative protein abundance of disomes compared to wild-type cells grown in rich medium (YEPD). Protein levels are shown in the order of the chromosomal location of their encoding genes. Protein levels of duplicated chromosomes are shown in red.

The following source data and figure supplements are available for figure 1:

**Source data 1**. TMT and SILAC data.

**Figure supplement 1**. SILAC and TMT mass spectrometry of wild-type vs wild-type cells.

**Figure supplement 2**. Transcriptome and proteome quantification of aneuploid yeast strains.

## Components of macromolecular complexes are significantly attenuated in disomic yeast strains

To characterize which genes are subject to dosage compensation, we performed a gene ontology (GO) analysis. Using a stringent cutoff of $\log_2$ ratio of 0.6 (3*SD) lower than the expected value of 1.0, we identified a total of 550 proteins encoded by the duplicated chromosomes that were significantly attenuated in disomic strains grown in rich medium (~21% of detected ORFs). Gene ontology analysis revealed that components of macromolecular complexes were significantly enriched (369 of 550, p value = 5.1 E−32), including all ribosomal subunits detected (113 of 550, p value = 6.2 E−43) (*Figure 2E*, *Figure 2—source data 2*). Furthermore, enrichment of subunits of macromolecular complexes among the attenuated proteins was observed for every disome analyzed (*Figure 2F*). Analysis of the identity of the attenuated proteins in disomes grown in selective medium and analyzed by SILAC showed similar results; a significant number of proteins encoded by genes located on the duplicated chromosomes were attenuated in every disomic strain and components of macromolecular complexes were significantly enriched (295 of 486, p value = 2.7 E−17, *Figure 2—figure supplement 2A*, *Figure 2—source data 2*). Importantly, there was a significant overlap between the two experiments; 287 proteins (57%) were significantly attenuated in both experiments (*Figure 2—figure supplement 2B*). Among these proteins, 76% are subunits of macromolecular complexes (218 of 287).

We previously quantified protein abundances in two disomic strains, disomes V and XIII, relative to wild-type cells (*Torres et al., 2010*). Consistent with our findings presented here, preliminary analyses indicated that subunits of macromolecular complexes were enriched among the dosage compensated genes (*Torres et al., 2010*). A subsequent quantitative proteomic study of five aneuploid yeasts obtained as progeny from triploid or pentaploid meioses found no evidence for the attenuation of proteins that form multi-subunit complexes (*Pavelka et al., 2010*). To better understand this discrepancy, we analyzed the protein measurements generated by *Pavelka et al. (2010)* (*Figure 3—figure supplement 1A*). In two strains, the ratios of protein levels of duplicated genes fit a sum of two normal distributions; one with a mean increase close to twofold, the other with significantly reduced mean close to zero ($R^2$'s = 0.94 and 0.92, *Figure 3—figure supplement 1B*). In the other three strains, the ratios of protein levels of duplicated genes fit normal distributions and show average increases significantly lower than the predicted twofold change (mean $\log_2$ ratios equal to 0.79, 0.77 and 0.65) (*Figure 3—figure supplement 1B*). Pavelka et al. showed that attenuation of protein levels of subunits of macromolecular complexes were small but not significant compared to proteins not found in complexes. Using the same list of complexes in *Pavelka et al. (2010)* (*Gavin et al., 2006*), we obtained similar results (*Figure 3—figure supplement 1C*). However, when we used a more up to date and manually curated list of subunits of macromolecular complexes (*Pu et al., 2009*), we found that statistically significant attenuation in proteins that form part of complexes takes places in all the strains (*Figure 3—figure supplement 1C*). Next, we focused on the identity of the most attenuated proteins and asked whether subunits of complexes were enriched among them. We used a stringent cutoff of $\log_2$ ratio of 0.6 lower than the mean increase in protein levels of the duplicate genes in each of the five aneuploid strains and found that between 23 and 38% of duplicated proteins were significantly attenuated (*Figure 3—figure supplement 1D*). Importantly, the attenuated proteins are enriched for subunits of macromolecular complexes in all five strains (*Figure 3—figure supplement 1D*). We

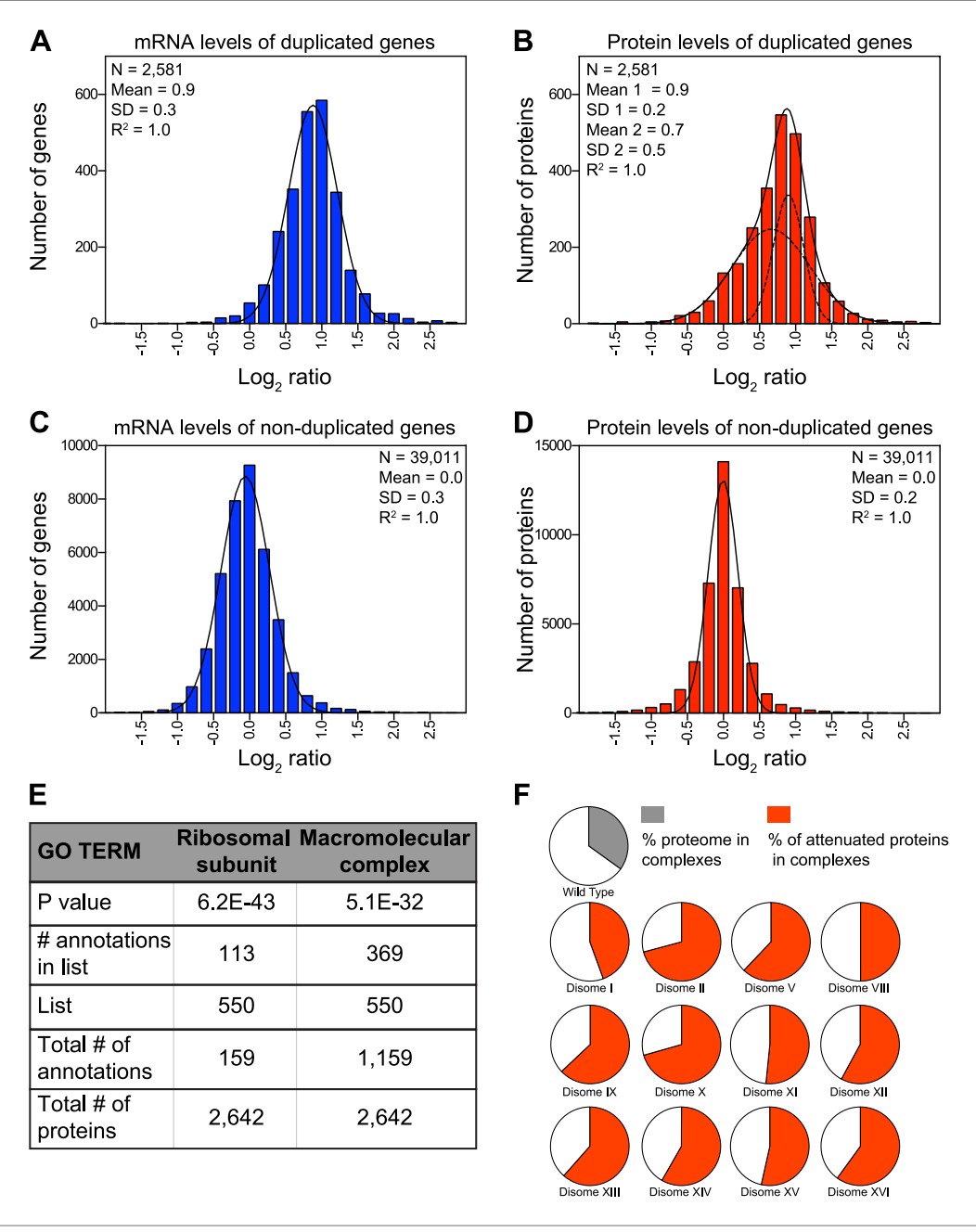

**Figure 2**. Attenuation of proteins encoded on duplicated chromosomes. (**A**) Histogram of the log$_2$ ratios of the relative mRNA levels of duplicated genes of 12 disomes relative to wild-type grown in YEPD medium. Fit to a normal distribution is shown (black line). (**B**) Histogram of the log$_2$ ratios of the relative protein levels of duplicated genes of 12 disomes relative to wild-type. Fit to the sum of two normal distributions is shown (black line). Fit of individual distributions are shown in dashed-line. (**C**) Histogram of the log$_2$ ratios of the relative mRNA levels of non-duplicated genes of 12 disomes relative to wild-type grown in YEPD medium. Fit to a normal distribution is shown (black line). (**D**) Histogram of the log$_2$ ratios of the relative protein levels of non-duplicated genes of 12 disomes relative to wild-type grown in YEPD medium. Fit to a normal distribution is shown (black line). (**E**) Gene Ontology enrichment analysis of 550 proteins encoded on duplicated genes that are significantly attenuated (log$_2$ ratios ≤ 0.4). (**F**) Pie chart representation of the relative number of all proteins predicted to form part of complexes in the yeast genome is shown in gray (33%). Pie chart representation of the relative number of proteins that are significantly attenuated and are part of macromolecular complexes in every disome are shown in red.

*Figure 2. Continued on next page*

*Figure 2. Continued*

The following source data and figure supplements are available for figure 2:

**Source data 1**. Gene expression data.
**Source data 2**. GO enrichment analysis.
**Figure supplement 1**. Attenuation of proteins encoded on duplicated chromosomes.
**Figure supplement 2**. Gene ontology analysis of attenuated proteins.

conclude that significant attenuation of subunits of macromolecular complexes is a general feature of aneuploid yeast strains.

## Posttranslational mechanisms are predominantly responsible for attenuation of protein levels

To determine the mechanisms that prevent an increase in protein levels despite increased gene dosage, we compared mRNA and protein levels of the attenuated genes in disomes grown in rich medium. Transcript levels of the attenuated proteins showed increases close to twofold and, unlike their protein products, showed no signs of compensation (*Figure 3A*). Strikingly, ribosomal genes encoded on duplicated chromosomes showed mean increases close to twofold in their mRNA levels while every ribosomal protein exhibited attenuation (*Figure 3B*). Similar results were obtained with cells grown in selective medium and analyzed by SILAC (*Figure 3—figure supplement 2*).

To investigate whether translational control mechanisms participate in the attenuation of protein levels, we performed ribosomal footprinting and SILAC-based proteome analysis of disome V and wild-type cells (*Figure 3C*, *Figure 3—source data 1*). In addition, we compared mRNA footprints of disome XVI previously published (*Thorburn et al., 2013*) to the proteome quantification of disome XVI grown in similar conditions. In both disomic strains, we found that duplicated genes, both attenuated and not attenuated at the protein level, show similar increases in ribosomal footprints (*Figure 3C,D*). While we cannot exclude the possibility that translational control may play a role in the attenuation of a small subset of genes, these results indicate that most of the duplicated genes are transcribed and translated. Our results show that dosage compensation is predominantly mediated by posttranslational mechanisms.

To test whether protein turnover pathways mediate the attenuation of duplicated genes, we performed TMT-based quantitative proteomics on wild-type cells and two aneuploid strains, disomes II and V, following inhibition of the proteasome and vacuolar degradation by addition of 100 μM MG132 and 10 mM chloroquine, respectively (note that the strains harbor a deletion in the gene encoding the efflux pump Pdr5 to increase the efficacy of MG132). We hypothesized that after very short times, 90 and 300 s, of protein turnover inhibition, only proteins with increased translation and that are rapidly degraded could show significant increases in abundance. These experiments revealed quantitative information for ~75% of all verified ORFs (*Figure 4A*, *Figure 4—source data 1*). As expected, very small changes in protein levels were detected in wild-type cells and the two disomes upon protein turnover inhibition (*Figure 4A*, *Figure 4—figure supplement 1A*). However, analysis of the average increase in protein levels per chromosome revealed that duplicated genes increased more than the rest of the genome (*Figure 4—figure supplement 1B*). Analysis of the identity of the duplicated genes revealed that attenuated proteins, which are enriched for subunits of complexes, account for most of the increases in protein levels (*Figure 4B,C*). Strikingly, individual proteins show increases between 4 to 20% in their levels upon inhibition of protein turnover in such short times (*Figure 4C*). These results provide direct evidence for protein degradation as being a mechanism for dosage compensation in aneuploid cells.

## Most macromolecular complexes harbor one or more subunits that are subject of dosage compensation

To determine which multi-subunit complexes were subject to subunit dosage compensation, we used a manually curated set of yeast protein complexes to assign complex status to all duplicated gene products (*Pu et al., 2009*). We found that more than half of the proteins designated as members of macromolecular complexes were significantly attenuated in the disomes grown in rich medium

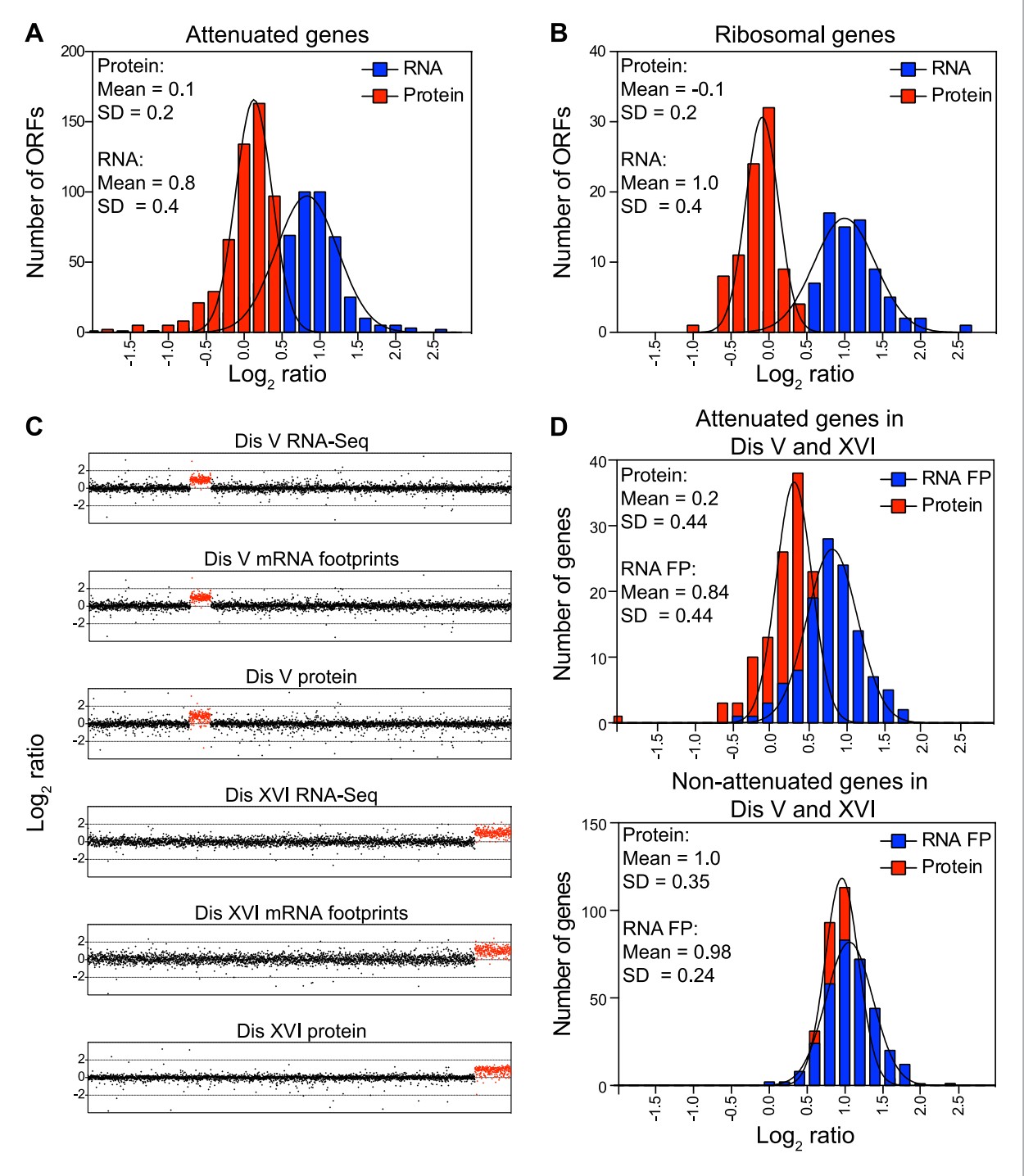

**Figure 3**. Attenuation takes place posttranslationally. (**A**) Histograms of the log$_2$ ratios of the relative mRNA (blue) and protein levels (red) of the 550 attenuated proteins from disomic cells grown in YEPD medium compared to wild-type. Fits to a normal distribution are shown (black lines). (**B**) Histograms of the log$_2$ ratios of the relative mRNA (blue) and protein levels (red) of 88 ribosomal protein genes. Fits to a normal distribution are shown (black lines). (**C**) The plots show the log$_2$ ratio of the relative mRNA levels, mRNA footprints and protein abundance of disomes V and XVI compared to wild-type cells. mRNA levels, mRNA footprints and protein levels are shown in the order of the chromosomal location of their encoding genes. Log$_2$ ratios of the duplicated chromosomes are shown in red. (**D**) Histograms of the log$_2$ ratios of the relative mRNA footprints (blue) and protein levels (red) of attenuated genes of disomes V and XVI compared to wild-type cells (top). Histograms of the log$_2$ ratios of the relative mRNA footprints (blue) and protein levels (red) of non-attenuated genes of disomes V and XVI compared to wild-type cells (bottom).

*Figure 3. Continued on next page*

*Figure 3. Continued*

The following source data and figure supplements are available for figure 3:

**Source data 1**. RNA-Seq and ribosome footprints of disome V.
**Figure supplement 1**. Analysis of proteome changes of meiotically generated aneuploid strains.
**Figure supplement 2**. Attenuation takes place posttranslationally in cells grown in selective medium.

(469 of 923 showed $\log_2$ ratios $\leq$ 0.6). Nonlinear regression analysis of the distribution of the $\log_2$ ratios of their levels relative to wild-type cells showed two populations, one of which encompassed the majority with a mean ratio of 0.46 (1.4-fold), a value significantly lower than the predicted increase of twofold that would be expected if protein levels scaled with gene copy number (*Figure 5A*). Similar results were obtained with cells analyzed by SILAC (*Figure 5—figure supplement 1A*). In contrast, proteins encoded by duplicated genes that are not found in complexes showed little attenuation (*Figure 5B*, *Figure 5—figure supplement 1B*). Nonetheless, levels of a small number of uncomplexed proteins were attenuated. To assess how these proteins contribute to the total attenuation, we analyzed their identity and the reproducibility of their attenuation in selective and rich media. We identified 88 proteins not known to function in complexes that were attenuated in both growth conditions. Gene ontology analysis did not reveal any significant enrichment for a particular function, cellular process or component. In fact, the biological function of 15 of 88 proteins is unknown (*Figure 5—figure supplement 2A*). Our analysis not only indicates that most of the proteome attenuation observed in aneuploid cells is caused by the attenuation of components of protein complexes but also leads us to estimate that about half of all cellular proteins found in complexes (469 of 923 detected proteins) are unstable and rapidly degraded unless they find their binding partners.

To identify which subunits of protein complexes are unstable when present in excess, we pooled quantitative information for subunits for individual complexes and calculated their average increase upon gene duplication. We limited our analysis to complexes for which quantitative information for three or more of their subunits was obtained in both TMT and SILAC datasets (*Figure 5—source data 1*). Analysis of 84 complexes is presented here (*Figure 5C*, *Figure 5—figure supplement 2B*, *Figure 5—source data 1*). Considering subunits that increase in levels by 1.5-fold or lower instead of the predicted to twofold ($\log_2$ ratio of 0.6 and lower), we found that 42 complexes showed attenuation in almost every subunit (average $\log_2$ ratios $\leq$ 0.6, *Figure 5C*, *Figure 5—figure supplement 2B*). Not surprisingly, subunits of the ribosome and the nucleosome were among the most attenuated proteins (*Figure 5C*, *Figure 5—figure supplement 2B*). The other 42 complexes showed average increases in their subunits higher than $\log_2$ ratios of 0.6 in one or both proteome datasets (*Figure 5C*, *Figure 5—figure supplement 2B*). Strikingly, with the exception of the trehalose-6-phosphate synthase/phosphatase complex (*Reinders et al., 1997*), every complex analyzed showed significant attenuation in at least one of its subunits (83 of 84).

Interestingly, nearly every complex also contains one or more subunits that are not attenuated, the ribosome and nucleosome being notable exceptions. A few examples include Arp3 of the heptameric Arp2/3 complex (*Robinson et al., 2001*), Mtw1 of the tetrameric MIND complex (*Maskell et al., 2010*), Gim5 of the hexameric prefoldin complex (*Vainberg et al., 1998*), Rpp1 of the nonameric ribonuclease P complex (*Chamberlain et al., 1998*), Prp19 of the octameric Prp19-associated complex (*Chen et al., 2002*), Orc5 of the hexameric origin recognition complex (*Bell and Stillman, 1992*), and, Ost1 of the nonameric oligosaccharyltransferase complex (*Spirig et al., 1997*). Examples of complexes with two stable subunits include Ccr4 and Not5 of the CCR4/NOT core complex which contains seven other subunits (*Chen et al., 2001*), Sec21 and Glo3 of the COPI complex which contains six other subunits (*Hosobuchi et al., 1992*), and Vma2 and Vma13 of the proton-transporting ATPase which contains 11 other subunits (*Kawasaki-Nishi et al., 2001*). Other stable subunits include proteins that can be found in more than one complex such as Rpb5 which is part of all three RNA polymerase I, II and III complexes (*Woychik et al., 1990*). While most cellular protein complexes appear to contain subunits that are highly unstable when present in excess, they may also require one or more stable subunits that serve as scaffolds for complex assembly.

To assess the reproducibility in attenuation of individual complex subunits, we compared their $\log_2$ ratios between cells grown in rich and selective medium, excluding the ribosome and nucleosome.

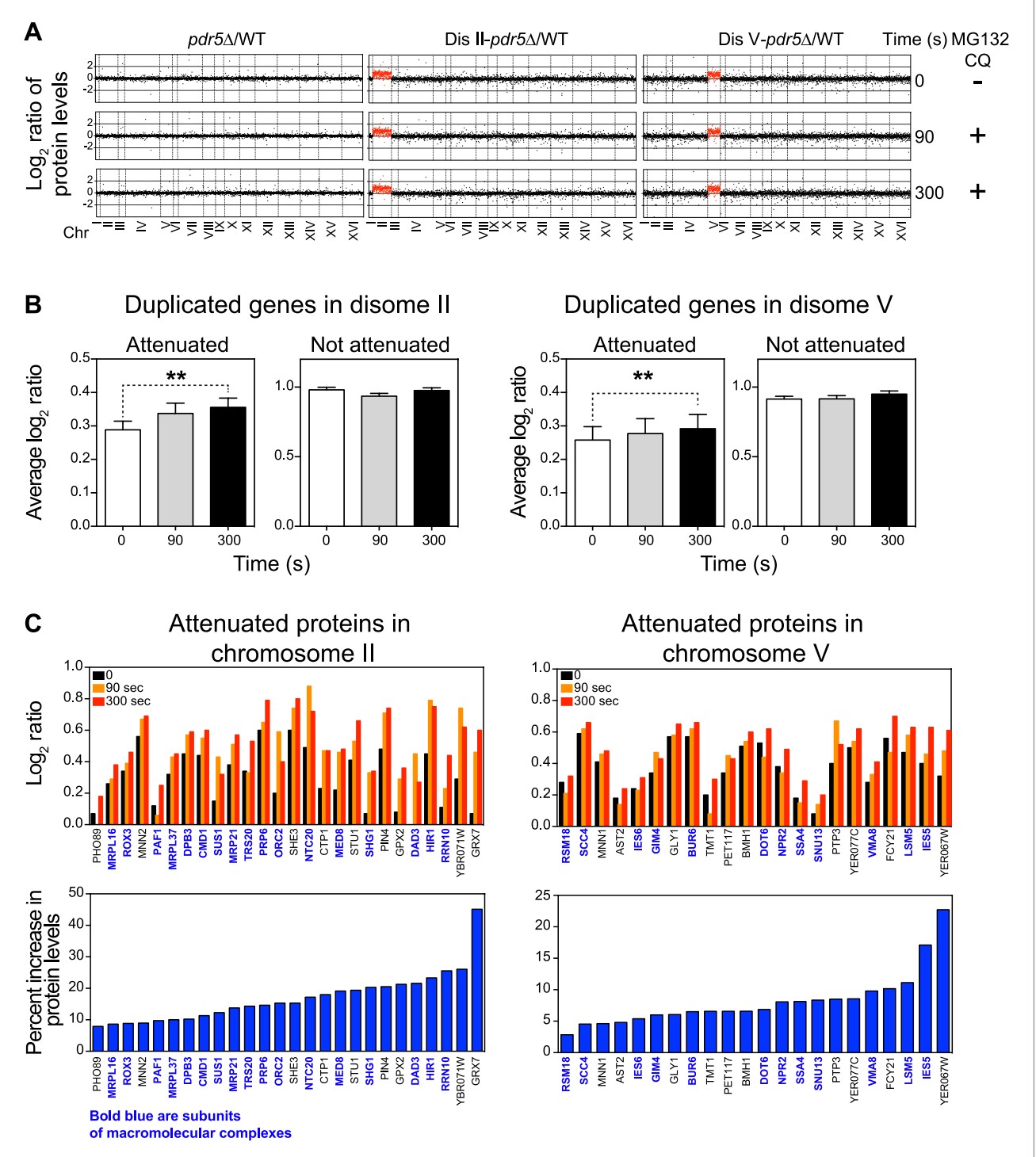

**Figure 4**. Inhibition of protein degradation leads to increases in levels of attenuated proteins. (**A**) The plots show the log₂ ratio of the relative protein abundance of disomes compared to wild-type cells and disomes II and V harboring the *PDR5* deletion compared to wild-type cells at 0, 90 and 300 s with 100 μM MG132 and 10 mM chloroquine. Protein levels are shown in the order of the chromosomal location of their encoding genes. Protein levels of duplicated chromosomes are shown in red. (**B**) Average log₂ ratios of attenuated and not attenuated products of duplicated genes in disome II (left) and disome V (right) upon inhibition of protein turnover for 0, 90 and 300 s (** denotes p values < 1E-3). (**C**) Examples of duplicated genes that are attenuated in disome II (left) and disome V (right) that show significant increases upon inhibition of protein degradation. Percent increases are shown below.

*Figure 4. Continued on next page*

*Figure 4. Continued*

The following source data and figure supplement are available for figure 4:

**Source data 1**. TMT proteome of WT, disome II and disome V after inhibition of protein turnover.
**Figure supplement 1**. Inhibition of protein degradation leads to increases in protein levels of duplicated genes.

*Figure 5D* shows the high correlation and reproducibility of the degree of attenuation of such proteins (Pearson r = 0.62) indicating that the effects described here are independent of growth conditions and quantification technique.

Attenuation of protein levels of duplicated genes could be a result of inherent instability of individual subunits. Alternatively, specific cellular responses to the presence of an extra chromosome could be a contributing factor. For example, the attenuation of ribosome subunit levels could be due to down-regulation of mRNA levels, which occurs as part of the environmental stress response (ESR) (*Gasch et al., 2000*). To distinguish between these mechanisms, we analyzed the effects of expressing ribosomal genes from centromeric plasmids on their protein levels. Western blot analysis of cells harboring plasmids with an extra copy of the ribosomal subunits *RPL1B*, *RPL3* or *RPL30* showed that protein levels of these genes did not increase with copy number (*Figure 5E*). In contrast, cells harboring plasmids with an extra copy of *ARP5* or *CDC28*, which encode proteins that are not attenuated, showed increased levels. Our results indicate that the protein attenuation of ribosomal subunits is at least in part driven by protein instability rather than aneuploidy-induced cellular responses.

## Aneuploidy induces protein responses through both transcriptional and posttranscriptional mechanisms

We previously identified a pattern of transcriptional changes in aneuploid yeast with similarity to the ESR (*Torres et al., 2007*). This change in gene expression leads to a corresponding change in protein levels (*Figure 6—figure supplement 1*). The ESR signature was also present in disomic yeast strains grown in rich medium, although with reduced intensity (*Figure 6—figure supplement 1*). We hypothesize that this reduced ESR is in part due to smaller differences in proliferation rates between disomic and wild-type cells grown in rich medium compared to selective medium (*Torres et al., 2007*). These results indicate that transcriptional responses to cellular stress and slow proliferation also affect the proteome of aneuploid cells.

To investigate whether additional protein responses are shared between aneuploid strains, we performed hierarchical clustering analysis of the protein changes for cells grown in rich medium after reducing the weight of the duplicated gene products ('Materials and methods'). We identified a novel signature of upregulated proteins in all of the disomes compared to wild-type cells (*Figure 6A*, *Figure 6—source data 1*). Here, we refer to this signature as the APS (aneuploidy-associated protein signature). Importantly, this protein signature was not observed in three independent wild-type/wild-type control experiments.

GO enrichment analysis of the APS revealed a group of proteins associated with cellular responses to oxidative stress, including thioredoxins Trx1 and Trx2, oxidoreductases Grx1 and Grx5, peroxiredoxins Ahp1 and Prx1, and the superoxide dismutase Sod1. In addition, the APS included proteins upregulated during oxidative stress such as the yeast orthologue of the translationally controlled tumor protein (p23) Tma19, the essential NTPase required for ribosome synthesis Fap7, the 3'-5'-exodeoxyribonuclease YBL055C, and the polyamine synthases Spe3 and Spe4 (*Gasch et al., 2000*; *Juhnke et al., 2000*; *Chattopadhyay et al., 2006*). These results indicate that aneuploid cells may be exposed to higher levels of intracellular reactive oxygen species (ROS) (see below). Another GO category of APS-enriched genes is 'metabolic processes', including functions such as amino acid biosynthesis and cellular bioenergetics. Interestingly, the intensity of the APS, measured as the average increase of its 92 proteins, correlated with the size of the additional chromosome (Pearson r = 0.62, *Figure 6B*), indicating that it may be a direct consequence of the cellular imbalances caused by the presence of the extra chromosome, rather than due to increased dosage of specific genes. In addition, we found that the APS is also present in aneuploid strains isolated from random meiosis and that its intensity also correlated with the size of the additional chromosomes in those aneuploid strains (*Figure 6—figure supplement 2*).

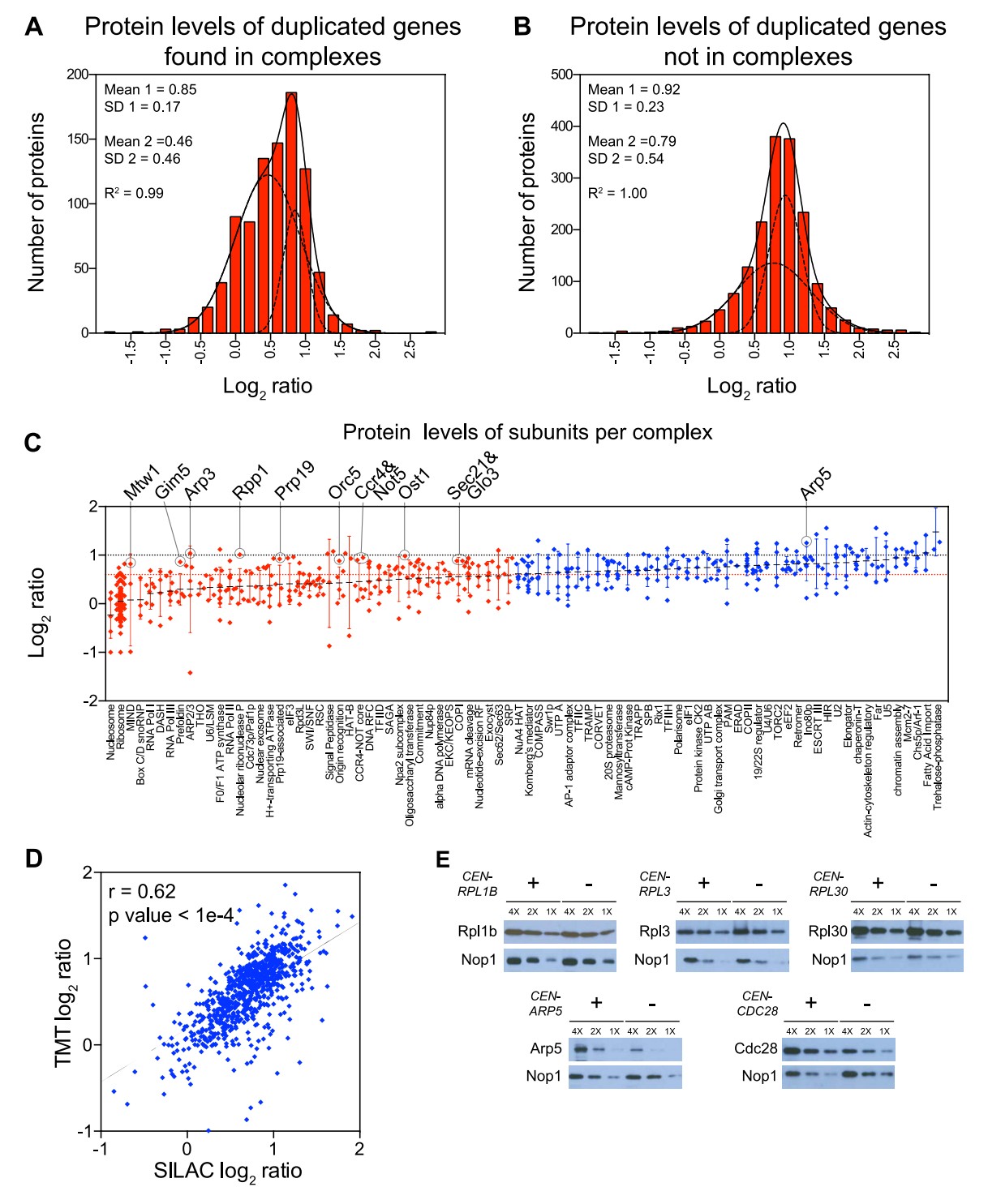

**Figure 5**. Subunits of macromolecular complexes accounts for most of the attenuation. (**A**) Histograms of the log₂ ratios of the relative protein levels of 923 duplicated genes found in complexes from disomic cells grown in YEPD medium compared to wild-type. Fits to a sum of two normal distributions are shown (black lines). (**B**) Histograms of the log₂ ratios of the relative protein levels of 1,658 duplicated genes not part of complexes from disomic cell grown in YEPD medium compared to wild-type. Fits to a sum of two normal distributions are shown (black lines). (**C**) Log₂ ratios of subunits of complexes when encoded in a duplicated chromosome relative to wild-type. Complexes that show significant attenuation mean of their subunits < 0.6 (dashed red line) are shown in red. (**D**) Comparison of the protein levels of subunits of complexes when present in a duplicated chromosome in disomic cells grown in
*Figure 5. Continued on next page*

*Figure 5. Continued*

YEPD vs synthetic medium. Pairwise comparison show a Pearson correlation coefficient (r) = 0.62. (**E**) Protein levels in wild-type cells or cells harboring a CEN plasmid containing a single copy of *RPL1B, RPL3, RPL30, ARP5* or *CDC28*.

The following source data and figure supplements are available for figure 5:

**Source data 1**. List of complexes analyzed.
**Figure supplement 1**. Subunits of macromolecular complexes accounts for most of the attenuation.
**Figure supplement 2**. GO term analysis of attenuated proteins not found in complexes.

Despite not finding a statistically significant enrichment for cellular processes associated with proteotoxic stress, we found several upregulated proteins involved in protein quality control pathways (***Figure 6—source data 1***). These include the Hsp90 regulators Sba1 and Hch1, the cis-trans peptide isomerases Cpr3, Fpr1 and Fpr3, and three proteins involved in ubiquitination including the ubiquitin-conjugating enzyme Ubc1, the ubiquitin interacting protein Duf1 and the ubiquitin-like protein Rub1. In addition, APS genes included several proteins involved in protein trafficking including Arc1, Sec53, Ric1, Vti1 and Ykt6. The upregulation of these proteins is consistent with increases in flux through protein folding, trafficking, and turnover machinery in aneuploid cells and will form the basis for future investigations. In support of a proposed need for increased protein degradation, we found that the average levels of proteasome subunits in all the disomic strains showed a small but significant increase compared to wild-type cells in almost every disomic strain independent of growth conditions (***Figure 6—figure supplement 3***).

Unexpectedly, the corresponding mRNA transcripts for most of the upregulated proteins were not increased. The average gene expression levels showed minimal changes (***Figure 6C***) indicating that the control of protein upregulation is posttranscriptional. Intriguingly, we did not detect the APS signature in cells grown in synthetic medium. We do not yet understand the reason for this difference but hypothesize that the larger changes in gene and protein expression due to the selective conditions may mask its detection.

## Intracellular reactive oxygen species are elevated in aneuploid yeast strains

Our proteome analysis revealed a response to oxidative stress in aneuploid yeast strains. To test whether this was due to defects in redox homeostasis, we compared the viability of wild-type cells and disomes in the presence of diamide or hydrogen peroxide ($H_2O_2$). We found that most disomes show hypersensitivity to the reactive oxygen species ROS-inducing agents diamide (1 mM) or $H_2O_2$ (3%) (***Figure 6D***, ***Figure 6—figure supplement 4A***). To investigate whether the mere presence of chromosome-size amounts of DNA was responsible for hypersensitivity to diamide or $H_2O_2$, we tested the viability of strains harboring a yeast artificial chromosome (YAC) varying in size containing human or mouse DNA. Cells harboring such YACs did not exhibit hypersensitivity to the ROS inducing agents (***Figure 6D***, ***Figure 6—figure supplement 4A***), indicating that the presence of the extra yeast genes and their products is responsible for the increased sensitivity to oxidative stress.

To test whether the upregulation of oxidative stress response proteins was due to increased levels of intracellular ROS, we measured ROS levels in the disomes during exponential growth using a fluorescent, ROS-sensitive dye, 5-(and-6)-chloromethyl-2′,7′-dichlorodihydrofluorescein diacetate (CM-$H_2$DCFDA) (***Figure 6E***, ***Figure 6—figure supplement 4B***). Basal levels of intracellular ROS were higher in most disomes compared to wild-type cells. ROS levels in cells harboring a YAC with human or mouse DNA did not show such increases (***Figure 6E***). Our results indicate that aneuploidy disrupts cellular redox homeostasis leading to the accumulation of intracellular ROS. Our data further suggest that aneuploid cells respond to these elevated ROS levels by maintaining higher protein levels of ROS scavengers such as thioredoxins and oxidoreductases.

## Loss of function of *UBP6* ameliorates changes in protein abundance in all disomic strains

Our previous studies identified loss of function mutations in the deubiquitinating enzyme *UBP6* as attenuating the proteomic changes of aneuploidy in two disomic strains (***Torres et al., 2010***). In one

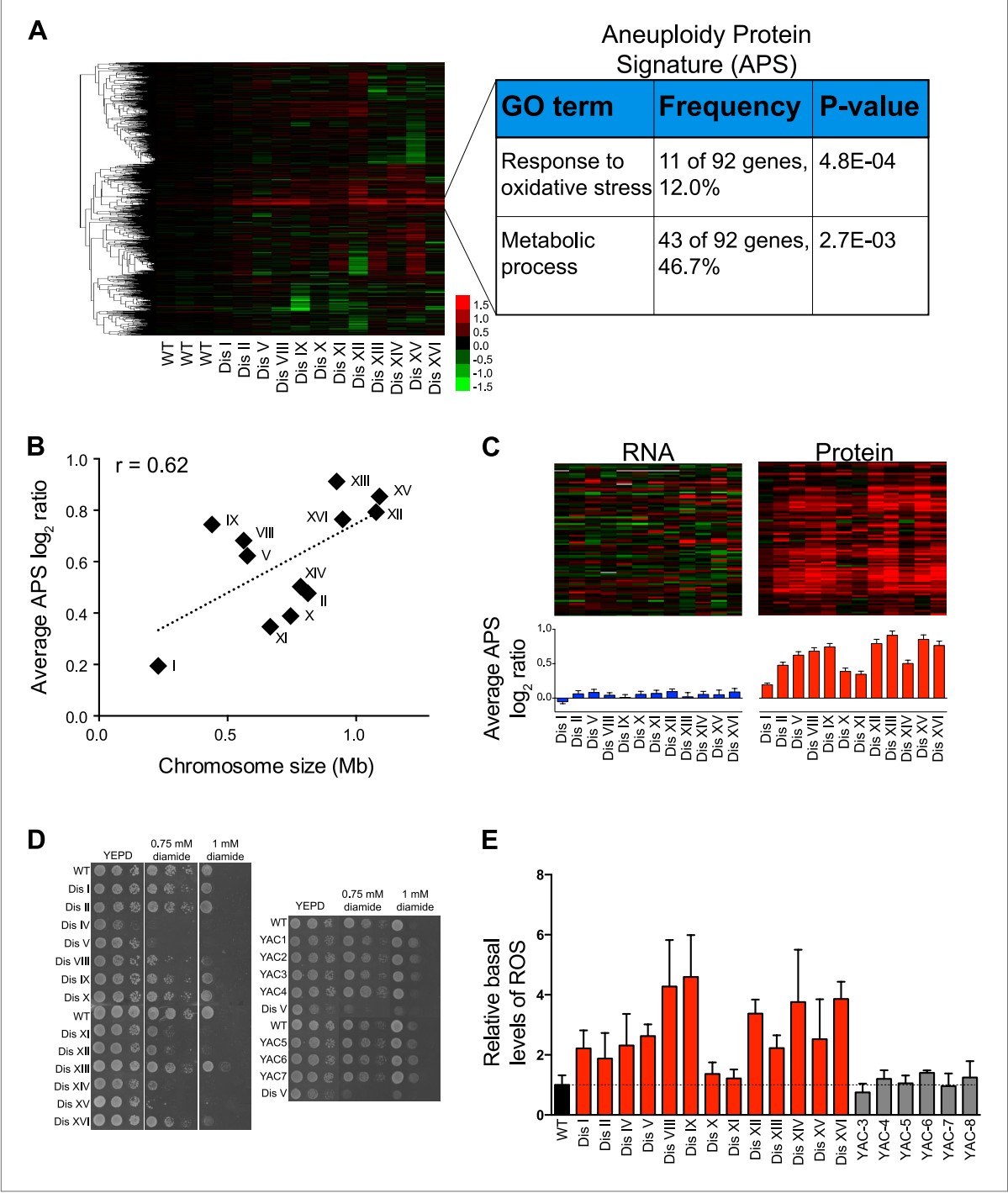

Figure 6. Identification of protein signature associated with aneuploidy. (A) Hierarchically clustered protein levels from strains grown in YEPD. Proteins encoded on duplicated chromosomes were down-weighted and all data were clustered using the program WCluster. Gene Ontology enrichment analysis of 92 proteins that are significantly upregulated in all 12 disomic strains is shown. We refer to this signature the aneuploidy-specific signature or APS. (B) Correlation of the average APS and chromosome size in the disomes. Linear fit is shown in dashed line. (C) Comparison of transcript (left) and protein levels (right) of the APS. Averaged gene (blue bars) or protein (red bars) expression of the APS of each disomic strain are shown below. Error bars represent SEM. (D) Proliferation capabilities of WT, disomes and cells harboring YACs on YEPD medium alone or in the presence of 0.75 or 1 mM diamide. (E) Relative ROS levels of the disomes grown in YEPD relative to wild-type cell. Error bars represent SD (n = 3).

*Figure 6. Continued on next page*

*Figure 6. Continued*

The following source data and figure supplements are available for figure 6:

**Source data 1**. List of the Aneuploidy-associated protein signature.
**Figure supplement 1**. Protein signatures associated with aneuploidy.
**Figure supplement 2**. APS in the meiotically generated aneuploid strains.
**Figure supplement 3**. Ribosome and proteasome levels in aneuploid cells.
**Figure supplement 4**. Proliferation capabilities of aneuploid cells in the presence of 3% $H_2O_2$.

strain (disome V) but not the other (disome XIII), this attenuation was associated with improved proliferative abilities. The studies described here show that aneuploidy profoundly impacts the proteome of all aneuploid yeast strains. We performed gene expression and proteomic analyses of 12 disomic strains harboring the deletion of *UBP6* (*ubp6Δ*). We measured both mRNA and protein levels for ~70–80% of all verified open reading frames (ORFs) in the disomes-*ubp6Δ* relative to wild-type cells (*Figure 7A*, *Figure 7—figure supplement 1A*, *Figure 7—source data 1*). Plots of the $\log_2$ ratios sorted by chromosomal position showed a strong correlation between mRNA and protein levels (*Figure 7A*, *Figure 7—figure supplement 1B*). While analysis of the $\log_2$ ratios of proteins encoded by non-duplicated genes showed a normal distribution (*Figure 7B*), $\log_2$ ratios of proteins encoded by duplicated genes fit a sum of two populations one of which was significantly attenuated (*Figure 7C*). The mRNA levels of these duplicated genes, however, showed an average increase of ~twofold with no signs of compensation (*Figure 7—figure supplement 1C*). Importantly, loss of *UBP6* did not further attenuate levels of proteins found to be dosage compensated in the disomic strains (Pearson r = 0.75, *Figure 7D,E*, *Figure 7—figure supplement 1D*). These results indicate that the *UBP6* deletion does not significantly alter attenuation of subunits of macromolecular complexes.

Next, we extended our analysis to all proteins whose levels were significantly altered in the disomes relative to wild-type cells regardless of their chromosomal origin. To this end, we binned proteins into three categories: upregulated ($\log_2$ ratio $\geq 0.4$), downregulated ($\log_2$ ratio $\leq -0.4$), and those that do not significantly change ($-0.4 < \log_2$ ratio $< 0.4$). We then compared the average of each category to the average change of the same proteins in the disomes lacking *UBP6*. We found that all 12 disomes showed significant attenuation in the levels of their most upregulated proteins upon loss of *UBP6* (*Figure 7F*). Importantly, analysis of the changes in gene expression of this set of proteins showed minimal attenuation of mRNA levels, indicating that the increased attenuation upon loss of *UBP6* is mediated posttranscriptionally (*Figure 7—figure supplement 2*). All disomes also showed significant increases of downregulated proteins upon the deletion of *UBP6* bringing their levels closer to wild-type cells (*Figure 7G*). The effect on downregulated proteins impacts fewer proteins than the upregulation (*Figure 7F,G*). These results indicate that protein attenuation upon loss of *UBP6* occurs in all aneuploid strains examined. Importantly, it affects both downregulated, and to a greater extent, upregulated proteins. Increased proteasomal degradation due to the loss of *UBP6* could be responsible for the downregulation of overexpressed genes. Which proteins are direct targets of Ubp6 remains to be investigated. So far, we found that deletion of *UBP6* in wild-type cells did not affect the half-life of six proteins, Fap7, Glc8, Bna5, Tma19, Trx1 and Trx2, whose increased abundance in disomic strains is attenuated when *UBP6* is deleted (*Figure 8—figure supplement 1*). It is thus possible that the *ubp6Δ*-mediated attenuation of overexpressed proteins in the disomes is indirect. How deletion of *UBP6* brings about an increase in the levels of proteins that are down-regulated in aneuploid strains is more difficult to explain and one must invoke indirect effects such as downregulation of negative regulators of gene expression.

## Loss of *UBP6* significantly attenuates the cellular responses to aneuploidy

Consistent with oxidative stress responsive protein levels being upregulated in aneuploid cells and attenuated upon loss of *UBP6*, we found that the APS was significantly reduced in the disomic yeast

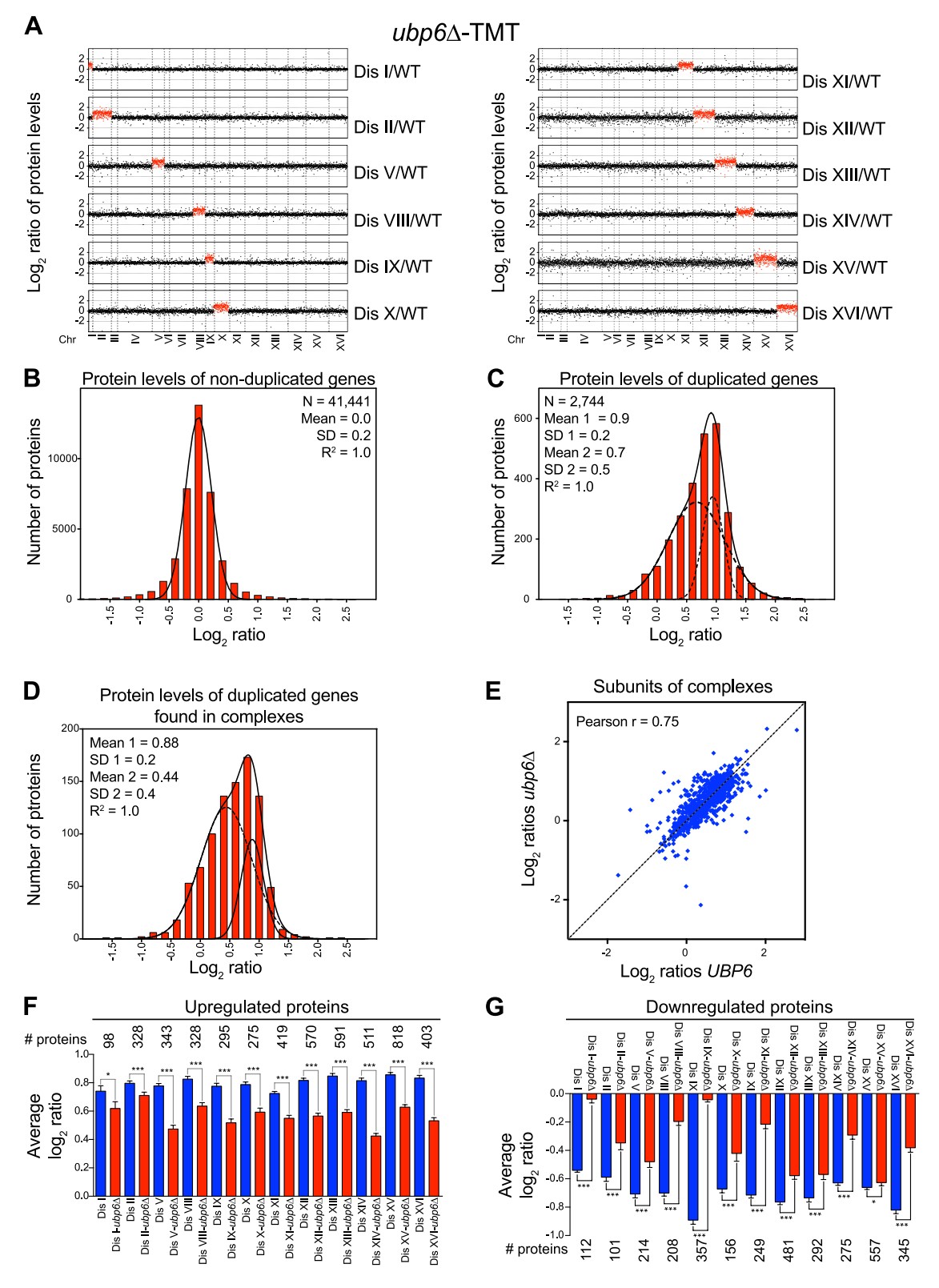

**Figure 7**. Loss of *UBP6* function preferentially affects proteins overproduced in disome V and disome XIII cells relative to wild-type. (**A**) The plots show the log₂ ratio of the relative protein abundance of disomes harboring the *UBP6* deletion compared to wild-type cells grown in YEPD. Protein levels are shown in the order of the chromosomal location of their encoding genes. Protein levels of duplicated chromosomes are shown in red. (**B**) Histogram of
*Figure 7. Continued on next page*

*Figure 7. Continued*

the log$_2$ ratios of the relative protein levels of non-duplicated genes of 12 disomes harboring the *UBP6* deletion relative to wild-type grown in YEPD medium. Fit to a normal distribution is shown (black line). (**C**) Histogram of the log$_2$ ratios of the relative protein levels of duplicated genes of 12 disomes harboring the *ubp6Δ* relative to wild-type grown in YEPD medium. Fit to a sum of two normal distributions is shown (black line). (**D**) Histograms of the log$_2$ ratios of the relative protein levels of duplicated genes found in complexes from disomic cells harboring the *UBP6* deletion grown in YEPD medium compared to wild-type. Fits to a sum of two normal distributions are shown (black lines). (**E**) Comparison of the protein levels of subunits of complexes when present in a duplicated chromosome in disomic cells vs disomes harboring the *UBP6* deletion grown in YEPD. Pairwise comparison show a Pearson correlation coefficient (r) = 0.75. (**F**) Average relative levels of the most upregulated proteins, log$_2$ ratios ≥ 0.4, in disomes-*UBP6* (blue) and disomes-*ubp6Δ* (red) compared to wild-type cells. Pair-wise *t* test was performed between disomes, * refers to p value = 0.01 and *** refers to p value < 1E-4. (**G**) Average relative levels of the most downregulated proteins, log$_2$ ratios ≤ −0.4, in disomes-*UBP6* (blue) and disomes-*ubp6Δ* (red) compared to wild-type cells. Pair-wise *t* test was performed between disomes, * refers to p value = 0.01 and *** refers to p value < 1E-4.

The following source data and figure supplements are available for figure 7:

**Source data 1**. Gene expression and proteome data of disomes-*ubp6Δ*.

**Figure supplement 1**. Analysis of mRNA and protein changes in disomes upon loss of *UBP6*.

**Figure supplement 2**. Protein levels of subunits of complexes and gene expression changes of the most up and downregulated genes in disomes-*ubp6Δ*.

strains lacking *UBP6* (**Figure 8A**). Interestingly, disome V, whose fitness is significantly improved upon deletion of *UBP6*, showed the strongest reduction in the APS. In contrast, we found that the ESR was not significantly affected in the disomic strains lacking *UBP6* (**Figure 8—figure supplement 2**). This is consistent with the fact that only 2 disomes, disome V and XI, show significant improvements in fitness when grown in rich medium (**Torres et al., 2010**). Our results indicate that loss of *UBP6* ameliorates protein responses associated with altered redox homeostasis and metabolism.

Amelioration of the APS upon *UBP6* loss of function (**Figure 8A**) suggests that the elevated intracellular ROS levels observed in the disomes may also be affected by the deletion. Indeed, loss of Upb6 function resulted in a significant decrease in the basal levels of intracellular ROS in 12 out of 13 disomic strains (**Figure 8B**). We were not able to test whether deletion of *UBP6* suppressed the diamide and H$_2$O$_2$ sensitivity of the disomic strains because deletion of *UPB6* itself causes sensitivity to these compounds. However, we were able to assess the effects of deleting *UBP6* on the overall fitness of the disomic strains. We previously found that loss of *UBP6* improved the fitness of 2 disomic strains when grown in YEPD medium and of 4 disomic strains when grown in selective medium (**Torres et al., 2010**). At high temperature the impact on fitness was more global. Deletion of *UBP6* suppressed the proliferation defect of 11 out of 13 disomic strains at 37°C (**Figure 8C**). Our results indicate that *UBP6* loss of function leads to lower intracellular ROS and ameliorates the APS in all disomic strains analyzed. Importantly, this suppression is also associated with an improvement in fitness in most disomic strains, particularly under proteotoxic stress conditions, suggesting that defects in redox homeostasis contribute to the proliferation defect of aneuploid strains.

## Discussion

### Aneuploidy alters the proteome

Our studies show that, in general, the acquisition of an extra chromosome translates into proportional increases in protein levels encoded on that chromosome. Despite the mechanisms that exist for dosage compensation of sex chromosomes, eukaryotic cells do not seem to have evolved mechanisms to silence genes upon the acquisition of an extra copy of an autosome. Therefore, a direct consequence of gaining an extra chromosome is increased flux through the transcription and translation machineries. However, the near comprehensive quantitative proteomic assessment of 12 disomic yeast strains described here reveals that levels of a sizeable portion of the proteome (~20%) do not scale with gene copy number. Most of the attenuated proteins are members of multi-protein complexes. Importantly, we show that attenuation of protein levels is mediated by posttranslational mechanisms, mostly by protein degradation. We hypothesize that these proteins only acquire a stably folded state when they are incorporated into their native complexes. Cells produce stoichiometric amounts of individual subunits of multiprotein complexes (**Li et al., 2014**). Approximately one third of

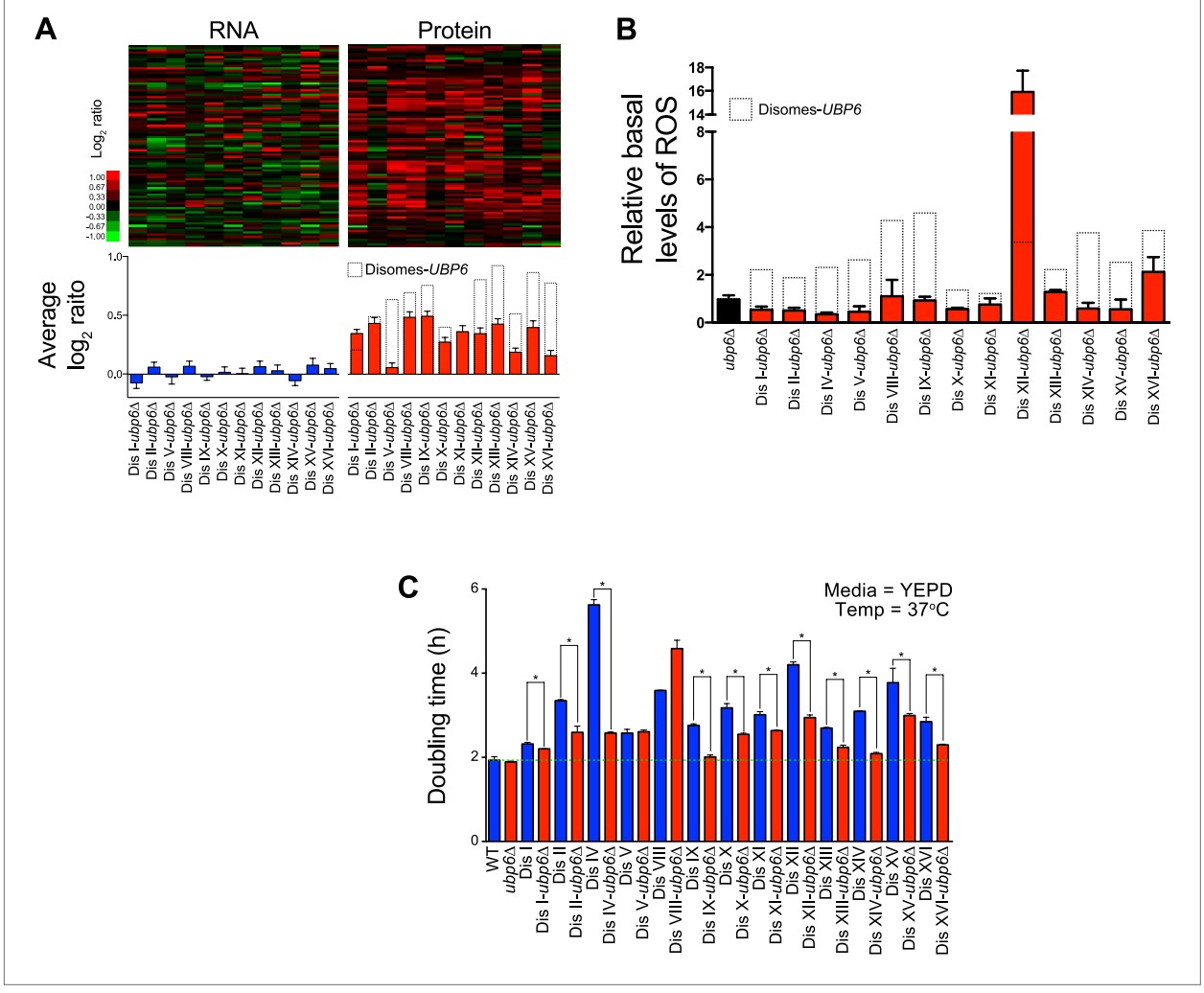

**Figure 8**. Loss of *UBP6* attenuates cellular responses to aneuploidy. (**A**) Comparison of transcript (left) and protein levels (right) of the APS in disomes-*ubp6Δ*. Averaged gene (blue bars) or protein (red bars) expression of the APS of each disomic strain are shown below. Error bars represent SEM. For comparison, dashed lines show the corresponding averages in disomes-*UBP6*. (**B**) Relative ROS levels of disomes-*ubp6Δ* grown in YEPD at 30°C. Error bars represent SD (n = 3). For comparison, dashed lines show the corresponding ROS levels in disomes-*UBP6*. (**C**) Doubling times of cells at 37°C. * refers to p-value < 0.05 (*t* test).

The following source data and figure supplements are available for figure 8:

**Source data 1**. List of strains utilized.

**Figure supplement 1**. Cyclohexidime chases of Ubiquitin, Trx1, Trx2, Bna5, Tma19, Fap7 and Glc8 in wild-type cells or cells harboring the *ubp6Δ*.

**Figure supplement 2**. ESR, ribosome and proteasome levels in disomes-*ubp6Δ*.

all yeast genes, which are randomly scattered in the genome, encode subunits of macromolecular complexes. Therefore irrespective of which chromosome is duplicated, significant protein attenuation must take place (***Figure 2F***). For example, the ribosome, which consists of 79 unstable proteins (encoded by 137 genes), is one of the most abundant multi-subunit complexes in the cell. It is estimated that 50% of all RNA Polymerase II transcription is devoted to the production of ribosomal proteins (***Warner, 1999***). Except for chromosomes I, III and VI, every yeast chromosome encodes several subunits, ranging from 5 in disome V to 19 in disome IV. Thus even a single excess chromosome leads to a substantial increase in transcription and translation of individual subunits without

upregulation of the total number of ribosomes. Our results indicate that the majority of excess subunits, not only of the ribosome but also of most complexes, are destined for degradation. Altogether, our results provide direct evidence that a major consequence of aneuploidy is an increased burden on the protein quality control pathways including protein degradation.

Cells have evolved several mechanisms that facilitate complex assembly, such as co-transcriptional regulation and dedicated chaperone systems that help stabilize unstable subunits to prevent their degradation. Our analysis indicates that two of the most stable and long-lived complexes in cells, the ribosome and the nucleosome, consist of subunits that may not exist for long unless assembled into their complex (*elBaradi et al., 1986*; *Gunjan and Verreault, 2003*; *Meeks-Wagner and Hartwell, 1986*; *Tsay et al., 1988*). Most other complexes show a large range of subunit stabilities. Remarkably, almost every complex analyzed in our study contains at least one attenuated subunit. Conversely, most macromolecular complexes with the exception of the nucleosome and ribosome contain at least one subunit that appears to be stable on its own. The existence of unfolded or partially folded unstable subunits may provide the necessary free energy to drive complex formation (*Kiefhaber et al., 2012*). Our results suggest that a stable scaffold protein may also be required for complex assembly. Deciphering the molecular mechanisms that dictate which subunits are degraded and which are not will significantly contribute to our understanding of the regulation of macromolecular complex formation.

## Cellular responses to aneuploidy

Aneuploidy hampers cellular proliferation and consistently elicits gene expression responses associated with slow proliferation and stress. Here, we showed that such gene expression responses affect the proteome content of cells. Paradoxically, one facet of the ESR is the downregulation of ribosomal protein genes leading to lower ribosome protein levels in the disomic strains compared to wild-type cells (*Figure 6—figure supplement 3B*). This is despite the apparent increase in total translation in cells harboring an extra chromosome. In addition, ribosomal footprinting analyses of disomic strains did not reveal any signs of impairment in translation efficiency. Therefore, the functional consequences of downregulation of ribosomes may be due to slower proliferation rates and may not affect the translational capacity of the cell. Nonetheless, increased translation and downregulation of ribosomal genes may provide the molecular explanation for the increased sensitivities of aneuploid cells to drugs that target the translational machinery.

In addition to the ESR-driven protein changes, we identified a novel aneuploidy-specific protein expression signature. This signature is present in all disomes analyzed and consists of 92 upregulated proteins involved in the regulation of redox homeostasis and metabolism. The upregulation of several of these proteins appears to occur in response to higher basal levels of intracellular ROS in the disomes. At present, we do not know the source of elevated intracellular ROS but our results indicate that disruption of protein homeostasis may be the culprit. Increased protein translation, folding and turnover create a high demand for ATP, which leads to the accumulation of ROS (*Gorrini et al., 2013*). In addition, endoplasmic reticulum (ER) stress due to increased protein folding could also contribute to ROS accumulation (*Tu and Weissman, 2002*). Another not mutually exclusive possibility is that altered metabolism due to upregulation of anabolic processes alters redox homeostasis in aneuploid cells (*Gorrini et al., 2013*). Consistently, several proteins involved in the biosynthesis of amino acids, nucleotides and lipids are upregulated in the disomes (*Figure 6—source data 1*). Lastly, our analysis indicates that the average increase in levels of the APS strongly correlates with the size of the extra chromosomes in the disomes, suggesting that this response may be a direct consequence of the acquisition of extra genes.

An unexpected finding in our studies is that the APS is not associated with increases in corresponding mRNA levels. Several potential mechanisms could mediate this response. Increased translation could be one reason. However, we found no detectable changes in translational efficiency of the APS genes in disomes V or XVI. We note that a more comprehensive ribosomal footprinting analysis would be necessary to reveal small but significant changes in translational control for a particular gene. It is also possible that the APS is the result of protein stabilization. Consistently, stabilization of proteins following transcriptional downregulation has been observed in cells exposed to mild oxidative stress over the course of several hours (*Vogel et al., 2011*). How stabilization occurs is not yet known but changes in posttranslational modifications such as phosphorylation are certainly one possibility. Most of the proteins of the APS have been shown to be ubiquitinated or phosphorylated (76 of 92,

*Figure 6—source data 1*). Because aneuploidy alters cellular metabolism, posttranslational modifications involving metabolites could also play a role. Interestingly, 34 of 92 APS proteins have been shown to be acetylated and/or succinylated in yeast (*Henriksen et al., 2012*; *Weinert et al., 2013*).

## Increased protein attenuation suppresses cellular responses to aneuploidy

Increased protein degradation mediated by the loss of function of *UBP6* suppresses several phenotypes associated with aneuploidy. Our results indicate that protein attenuation upon loss of *UBP6* occurs independently of the identity of the extra chromosome and that it affects both down and upregulated proteins, although the latter to a greater extent. We hypothesize that deletion of *UBP6* directly affects protein degradation and/or ameliorates protein responses indirectly by suppressing aneuploidy-associated phenotypes. Another possibility is that loss of *UBP6* directly leads to increased degradation of a few transcriptional regulators thereby affecting the levels of both down and upregulated proteins.

Loss of *UBP6* suppresses the sensitivity to high temperature exhibited by most disomic strains. In addition, we found that loss of *UBP6* suppresses the APS and reduces elevated basal levels of reactive oxygen species in most disomes. Importantly, analysis of the attenuated proteins revealed that attenuation of subunits of complexes was not increased; thus *UBP6* does not appear to be involved in their degradation. These findings also suggest that *UBP6* substrates are enriched for proteins involved in stress responses. Whether the most attenuated proteins in the disomes upon *UBP6* loss are direct targets of Ubp6's deubiquitinating activity is not clear and requires further investigation. Nonetheless, loss of *UBP6* leads to the clearance of protein aggregates in aneuploid cells (*Oromendia et al., 2012*). This raises the possibility that the removal of protein aggregates could contribute to the beneficial effects of *UBP6* deletion. Because protein aggregates sequester numerous proteins with essential cellular functions (*Olzscha et al., 2011*), their removal upon loss of *UBP6* may release sequestered proteins and could account for increases of downregulated proteins in the disomes. Alternatively, protein aggregates may consist of other metastable proteins simply as a consequence of impaired folding and/or chaperone activity and attenuation in protein abundance mediated by *UBP6* alleviates such stress. Establishing which proteins are direct targets of Ubp6 will help us understand the molecular mechanisms by which its loss of function suppresses aneuploidy-associated phenotypes.

## Implications for human disease

Our studies revealed that aneuploidy leads to higher levels of intracellular ROS. This increase in ROS may in part be responsible for the genomic instability observed in aneuploid cells (*Sheltzer et al., 2011*). Consistent with this, aneuploid mouse embryonic fibroblasts as well as most cancer cells are characterized by high levels of reactive oxygen species (*Li et al., 2010*; *Gorrini et al., 2013*). Unexpectedly, cells respond to increases in ROS by maintaining elevated levels of ROS scavenger proteins by posttranscriptional mechanisms. Our studies raise an important question: how do cancer cells exploit posttranscriptional mechanisms to alter protein levels and respond to intrinsic genomic alterations? Quantification of the cancer proteome remains a formidable challenge; but such efforts hold significant potential to reveal novel insights into the mechanisms by which cancer cells thrive despite their unbalanced genome. Deciphering such mechanisms could significantly impact our understanding of tumorigenesis.

Finally, our studies indicate that attenuation of proteome changes and removal of protein aggregates significantly ameliorates the detrimental effects of aneuploidy. Aneuploidy causes Down syndrome and is thought to play an active role in neurodegenerative diseases (*Siegel and Amon, 2012*). Our studies indicate that targeting genes in the protein degradation pathway, such as *UBP6*, holds significant potential to ameliorate the detrimental consequences of aneuploidy in humans. This opens the window for the design of novel approaches to improve the symptoms of Down patients and prevent or delay the onset of Alzheimer's or Huntington's disease.

## Material and methods

### Yeast strains and growth conditions

All stains are derivatives of W303 (E187) and are listed in *Figure 8—source data 1*. CEN-plasmids were isolated from the MOBY collection and introduced into wild-type cells by transformation. Gene expression analysis was performed as described in *Torres et al. (2007)* and is available in *Figure 2—source data 1*. All aneuploid strains used in this study were subjected to comparative genomic hybridization (CGH) to ensure that the additional chromosome was present in its entirety.

## Rationale for performing proteomic analysis of 12 out of 16 possible disomic yeast strains

Disomic yeast strains were generated by a chromosome transfer strategy described in *Torres et al. (2007)*. Cells disomic for chromosomes III and VII were not obtained because the MAT locus and the CYH2 locus located on chromosome III and VII, respectively, are required for selection steps during chromosome transfer procedure. Cell disomic for chromosome VI could not be generated as two copies of ACT1 and TUB2 seem to cause lethally (*Anders et al., 2009*). Cells disomic for chromosome IV, the largest chromosome in yeast, were not analyzed because they show poor cell viability (*Torres et al., 2007*).

## Growth of cells for SILAC analysis

Cells were grown overnight at 30°C in selective medium (-Lys-His+G418) in the presence of 'light' or 'heavy' lysine (100 mg/ml). Batch cultures were diluted to $OD_{600nm} = 0.2$ the next day and harvested once they reached an $OD_{600nm} = 1.0$.

## Distribution analysis of protein log$_2$ ratios

Analysis of the log$_2$ ratios was performed utilizing the PRISM software (v6.0). Pearson mode skewness was calculated as follows: (median − mean)/SD. Scatterplots and their correlation values (Pearson r) were also calculated with the PRISM software.

## Identification of APS

Hierarchical clustering was performed using the program WCluster (http://function.princeton.edu/WCluster/). WCluster takes both a data table and a weight table to allow individual measurements to be differentially considered by the clustering algorithm. Protein expression data were clustered by a Pearson correlation metric with equal weighting given to all data, or with no weight given to genes on the duplicated chromosomes.

## ROS measurements

Cells were grown overnight at 30°C in selective medium (-His+G418). Batch cultures were diluted to $OD_{600nm} = 0.2$ into YEPD medium the next day. Once they reached an $OD_{600nm} = 1.0$, cells were transferred to PBS buffer and incubated with 1 μM CM-H$_2$DCFDA at 30°C for 60 min. Excess dye was washed three times and cell fluorescence was analyzed by FACS.

## Polyribosome profile analysis

Polysomes were prepared as described (*Clarkson et al., 2010*). Briefly, 250-ml cultures were grown in YEPD at 30°C to an $OD_{600nm}$ of 0.5. Cycloheximide was added to a final concentration of 0.1 mg/ml for 3 min. Cells were pelleted by centrifugation and lysed by vortexing with zirconia/silica beads in 1× PLB (20 mM 4-(2-hydroxyethyl)-1-piperazineethanesulfonic acid–KOH, pH 7.4, 2 mM magnesium acetate, 100 mM potassium acetate, 0.1 mg/ml cycloheximide, 3 mM dithiothreitol [DTT]) and treated with RNasin Plus RNase inhibitor (Promega, Fitchburg, WI). Lysates were clarified by centrifugation, and 25 A260 units were resolved on 11-ml linear 10–50% sucrose gradients in 1× PLB by centrifugation in a Beckman SW41 rotor (Beckman Coulter, Indianapolis, IN) for 3 hr at 35,000 rpm.

## Mass spectrometry sample preparation

For SILAC experiments, cells grown in heavy and light media were mixed in equal numbers and lysed by bead beating in a buffer containing 8 M urea, 75 mM NaCl, 50 mM Tris-Cl, pH 8.2, and a protease inhibitor cocktail (complete mini, Roche, Germany) using three cycles of 90 s separated by three minute incubation on ice. The lysates were cleared of unlysed cells and insoluble material by centrifugation at 14,000×*g* for 15 min at 4°C. Protein concentrations were determined by a dye binding assay (Bio-Rad, Hercules, CA). Disulfide bonds were reduced by adding dithiothreitol (Sigma, St. Louis, MO) to a final concentration of 5 mM and incubating at room temperature for 40 min. Reduced cysteines were alkylated by the addition of iodoacetamide to 15 mM and incubation for 40 min in the dark at room temperature. Alkylation was quenched with an additional 10 mM dithiothreitol. Lysates were diluted 2.5-fold with Tris–HCl, pH 8.8 (25 mM final concentration). Lysyl endopeptidase (lysC, Wako, Richmond, VA) was added to a final concentration of 10 ng/ml and digests were allowed to proceed overnight at room temperature with gentle agitation. Digestion was stopped by the addition of formic acid (FA) to a final concentration of 1% and precipitates were removed by centrifugation at 14,000×*g* for 3 min. The supernatants were applied to pre-equilibrated Sep-Pak tC18 columns (Waters, Milford, MA)

and the columns were washed with 1% formic acid. Bound peptides were eluted with 70% acetonitrile (ACN), 1% FA and lyophilized.

## Growth of cells for TMT analysis

Cells were grown overnight at 30°C in selective medium (-His+G418). Batch cultures were diluted to $OD_{600nm} = 0.2$ into YEPD medium the next day and harvested once they reached an $OD_{600nm} = 1.0$.

## TMT-labeling

100 µg total peptide from each strain was resuspended in 100 µl of 0.2 M Hepes (pH 8.5). TMT six-plex reagents (0.8 mg per vial) (Thermo Fisher, Rockford, IL) were resuspended in 41 µl of anhydrous ACN and 10 µl of each reagent was added to each sample. Reactions were allowed to proceed at room temperature for 1 hr, after which they were quenched by the addition of 8 µl of 5% hydroxylamine for 15 min and then acidified by the addition of 16 µl neat FA. Reaction products from all six differentially labeled samples were combined and 1 ml of 1% FA was added before desalting on a 200-mg tC18 Sep-Pak. Eluted peptides were dried in a SpeedVac and stored at −20°C.

## Peptide fractionation

SILAC Peptides were separated by strong cation exchange (SCX) chromatography as described previously (*Villen and Gygi, 2008*) with minor changes. Briefly, 500 µg of an equal mix of heavy and light peptides, were resuspended in 250 µl of SCX buffer A (7 mM KH2PO4, pH 2.65, 30% ACN). Peptides were separated on a 4.6 mm × 200 mm polysulfoethyl aspartamide column (5 µm particles; 200 Å pores; PolyLC) using a 36 min gradient from 0% to 50% buffer B (7 mM KH2PO4, pH 2.65, 30% ACN, 350 mM KCl) at a flow rate of 1 ml/min. Fractions were collected every 1.5 min, freeze-dried, resuspended in 1% FA, and desalted using self-packed C18 STAGE-tips (*Rappsilber et al., 2003*). Peptides were eluted into glass inserts with 70% ACN/1% FA, dried, and resuspended in 100 µl of 5% FA.

TMT-labeled peptides were separated by high-pH reverse-phase HPLC (*Wang et al., 2011*). 600 µg of six-plex labeled peptides were resuspended in 250 µl buffer A (5% ACN, 10 mM $NH_4HCO_3$, pH 8) and separated on a 4.6 mm × 250 mm 300Extend-C18, 5 µm column (Agilent) using a 50 min gradient from 18% to 38% buffer B (90% acn, 10 mM $NH_4HCO_3$, pH 8) at a flow rate of 0.8 ml/min. Fractions were collected over 45 min at 28 s intervals beginning 5 min after the start of the gradient in a 96-well plate and lyophilized. Fractions were resuspended in 30 µl 1% FA and pooled into 12 samples of four fractions each (only 48 of 96 fractions were used) by combining fractions 1/25/49/73, 3/27/51/75, 5/29/53/77, 7/31/55/79, 9/33/57/81, 11/35/59/83, 14/38/62/86, 16/40/64/88, 18/42/66/90, 20/44/68/92, 22/46/70/94, 24/48/72/96 into glass vial inserts. This pooling strategy serves to minimize peptide overlap between fractions. The pooled samples were dried down and resuspended in 25 µl of 5% FA.

## LC-MS/MS analysis

For SILAC experiments, 2–4 µl (~1–3 µg) of each SCX fraction was analyzed by LC-MS/MS on a LTQ-Orbitrap, LTQ-Orbitrap Discovery, or LTQ-Velos hybrid linear ion trap (ThermoFisher). Between 17 and 25 fractions were analyzed for each experiment. In some cases, depending on separation quality and/or instrument performance, samples were run twice pooling both sets of data. Peptides were introduced into the mass spectrometer by nano-electrospray as they eluted off a self-packed 18 cm, 100 µm (ID) reverse-phase column packed with either 5 µm or 3 µm, 200 Å pore size, Maccel C18 AQ resin (The Nest Group, Southborough, MA). Peptides were separated using a 95 min or 65 min (Velos, Germany) gradient of 5–27% buffer B (97% ACN, 0.125% FA) with an in-column flow rate of 0.3–0.5 µl/min. For each scan cycle, one high mass resolution full MS scan was acquired in the Orbitrap mass analyzer and up to 10 or 20 (Velos) parent ions were chosen based on their intensity for collision induced dissociation (CID) and MS/MS fragment ion scans at low mass resolution in the linear ion trap. Dynamic exclusion was enabled to exclude ions that had already been selected for MS/MS in the previous 60 s. Ions with a charge of +1 and those whose charge state could not be assigned were also excluded. All scans were collected in centroid mode.

For TMT experiments, 2–4 µl of each fraction was analyzed on a LTQ Orbitrap Velos mass spectrometer (Thermo Fisher Scientific) equipped with an Accela 600 quaternary pump (Thermo Fisher Scientific) and a Famos Microautosampler (LC Packings, Netherlands). Peptides were separated with a gradient of 6–24% ACN in 0.125% FA over 150 min and detected using a data-dependent Top10-MS2/MS3 'multi-notch' method (*Ting et al., 2011*; *McAlister et al., 2014*). For each cycle, one full MS scan was

acquired in the Orbitrap at a resolution of 30,000 or 60,000 at m/z = 400 with automatic gain control (AGC) target of $2 \times 10^6$. Each full scan was followed by the selection of the most intense ions, up to 10, for collision-induced dissociation (CID) and MS2 analysis in the linear ion trap for peptide identification and subsequent higher-energy collisional dissociation (HCD) and MS3 analysis in the Orbitrap for quantification of the TMT reporter ions. AGC targets of $4 \times 10^3$ and $2 \times 10^4$ were used for MS2 and MS3 scans, respectively. Ions selected for MS2 analysis were excluded from reanalysis for 90 s. Ions with +1 or unassigned charge were also excluded from analysis. A single MS3 scan was performed for each MS2 scan selecting the most intense ions from the MS2 for fragmentation in the HCD cell. The resultant fragment ions were detected in the orbitrap at a resolution of 7500. Maximum ion accumulation times were 1000 ms for each full MS scan, 150 ms for MS2 scans, and 250 ms for MS3 scans.

## Database searching and filtering

MS/MS spectra were matched to peptide sequences using SEQUEST v.28 (rev. 13) (*Eng et al., 1994*) and a composite database containing the translated sequences of all predicted open reading frames of *Saccharomyces cerevisiae* (http://downloads.yeastgenome.org) and its reversed complement. Search parameters allowed for two missed cleavages, a mass tolerance of 20 ppm, a static modification of 57.02146 Da (carboxyamidomethylation) on cysteine, and dynamic modifications of 15.99491 Da (oxidation) on methionine. For SILAC samples, parameters also included a dynamic modification of 8.01420 Da on lysine. For TMT samples a static modification of 229.16293 Da on peptide amino termini and lysines was added.

Peptide spectral matches were filtered to 1% FDR using the target-decoy strategy (*Elias and Gygi, 2007*) combined with linear discriminant analysis (LDA) (*Huttlin et al., 2010*) using the SEQUEST Xcorr and $\Delta$Cn' scores, precursor mass error, observed ion charge state, and the number of missed cleavages. LDA models were calculated for each LC-MS/MS run with peptide matches to forward and reversed protein sequences as positive and negative training data. The data were further filtered to control protein-level FDRs. Protein scores were derived from the product of all LDA peptide probabilities, sorted by rank, and filtered to 1% FDR. The FDR of the remaining peptides fell markedly after protein filtering. Further filtering based on the quality of quantitative measurements (see below) resulted in a final protein FDR < 1% for all experiments. Remaining peptide matches to the decoy database as well as contaminating proteins (e.g., human keratins) were removed from the final data set.

## Peptide quantification

SILAC ratios were calculated automatically using the VISTA program (*Bakalarski et al., 2008*), requiring either a minimum signal-to-noise ratio ≥ 2 for both heavy and light or signal-to-noise ≥ 5 for one of the two.

For TMT experiments raw reporter ion intensities were denormalized by multiplying with the ion accumulation times for each MS3 scan and corrected for isotopic overlap between reporter ions by using empirically derived values. We required each peptide to have denormalized reporter ion intensities ≥ 20 for the zero time point and at least four of six TMT channels.

## Protein quantification

In all experiments, protein ratios were normalized to account for small variations in cell mixing by recentering the $\log_2$ protein abundance ratio distributions over zero using the assumption that most proteins are present at a one-to-one ratio. Proteins coded on the duplicated chromosomes, which are more abundant in the disomes were excluded when calculating this normalization factor. Protein ratios from the SILAC experiment were calculated as described (*Torres et al., 2010*) using the median $\log_2$ ratio of all peptides for each protein. For TMT experiments, relative protein abundances were calculated as the weighted average of all peptides from each protein using the ratio of the summed reporter ion intensities in each channel. Ratios for both experiments were $\log_2$-transformed for all subsequent analysis.

## Gene expression arrays

Total RNA was isolated from cells frozen on filters. Filters were incubated for 1 hr at 65°C in lysis buffer (10 mM EDTA, 0.5% SDS, and 10 mM Tris, pH 7.5) and acid phenol. The aqueous phase was further extracted twice with an equal volume of chloroform using phase lock gel (Eppendorf, Germany). Total RNA was then ethanol precipitated and further purified over RNeasy columns (Qiagen, Germany). RNA quality was checked using the Bioanalyzer RNA Nano kit, and 325 ng was used for microarray labeling with the Agilent Low RNA Input Fluorescent Linear Amplification Kit. Reactions were performed as directed except using half the recommended reaction volume and one quarter the

recommended Cy-CTP amount. Dye incorporation and yield were measured with a Nanodrop spectrophotometer. Equal amounts of differentially labeled control and sample cRNA were combined such that each sample contained at least 2.5 pmol dye. Samples were mixed with control targets, fragmented, combined with hybridization buffer, and hybridized to a microarray consisting of 60mer probes for each yeast open reading frame (Agilent). Microarrays were rotated at 60°C for 17 hr in a hybridization oven (Agilent, Santa Clara, CA). Arrays were then washed according to the Agilent SSPE wash protocol, and scanned on an Agilent scanner. The image was processed using the default settings with Agilent Feature Extraction software. All data analysis was performed using the resulting $\log_2$ ratio data, and filtered for spots called as significantly over background in at least one channel.

## Accession numbers

mRNA expression data for cells grown in synthetic medium were obtained from the GEO database with accession number GSE7812. mRNA expression data for cells grown in YEPD medium have been deposited at the GEO database with accession number GSE55166. The mass spectrometry proteomics data have been deposited to the ProteomeXchange Consortium via the PRIDE partner repository with the dataset identifier PXD001019 (*Vizcaino et al., 2014*). Data available from the Dryad Digital Repository: http://dx.doi.org/10.5061/dryad.65364 (Dephoure et al., 2014)

## Acknowledgements

We are thankful to Gloria Brar and Jonathan Weissman for their help with the mRNA footprinting analysis. We are grateful to Johnathan R Warner for reagents. This research was supported by the Richard and Susan Smith Family Foundation and the Searle Scholars Program to ET and a grant by the National Institutes of Health (GM056800) to AA. AA is also an investigator of the Howard Hughes Medical Institute.

## Additional information

### Funding

| Funder | Grant reference number | Author |
| --- | --- | --- |
| Richard and Susan Smith Family Foundation | s67400000023429 | Eduardo M Torres |
| Searle Scholars Program | 13-ssp-268 | Eduardo M Torres |
| National Institutes of Health | GM056800 | Angelika Amon |
| Howard Hughes Medical Institute | | Angelika Amon |

The funders had no role in study design, data collection and interpretation, or the decision to submit the work for publication.

### Author contributions

ND, EMT, Conception and design, Acquisition of data, Analysis and interpretation of data, Drafting or revising the article; SH, CO'S, SED, Acquisition of data, Analysis and interpretation of data; SPG, Conception and design, Analysis and interpretation of data; AA, Conception and design, Drafting or revising the article

## Additional files

### Major datasets

The following datasets were generated:

| Author(s) | Year | Dataset title | Dataset ID and/or URL | Database, license, and accessibility information |
| --- | --- | --- | --- | --- |
| Dephoure N, Hwang S, O'Sullivan C, Dodgson SE, Gygi SP, Amon A, Torres EM | 2014 | Quantitative Proteomic Analysis Reveals Posttranslational Responses to Aneuploidy in Yeast | PXD001019; http://proteomecentral.proteomexchange.org/cgi/GetDataset | Publicly available at ProtemeXchange (http://www.proteomexchange.org/). |

| Dephoure N, Hwang S, O'Sullivan C, Dodgson SE, Gygi SP, Amon A, Torres EM | 2014 | Quantitative Proteomic Analysis Reveals Posttranslational Responses to Aneuploidy in Yeast | GSE55166; http://www.ncbi.nlm.nih.gov/geo/query/acc.cgi?acc=+GSE55166 | Publicly available at the Gene Expression Omnibus (http://www.ncbi.nlm.nih.gov/geo/). |
| Dephoure N, Hwang S, O'Sullivan C, Dodgson SE, Gygi SP, Amon A, Torres EM | 2014 | Data from: Quantitative Proteomic Analysis Reveals Posttranslational Responses to Aneuploidy in Yeast | http://dx.doi.org/10.5061/dryad.65364 | Available at Dryad Digital Repository under a CC0 Public Domain Dedication. |

The following previously published dataset was used:

| Author(s) | Year | Dataset title | Dataset ID and/or URL | Database, license, and accessibility information |
| --- | --- | --- | --- | --- |
| Pavelka N, Rancati G, Zhu J, Bradford WD, Saraf A, Florens L, Sanderson BW, Hattem GL, Li R | 2010 | Aneuploidy confers quantitative proteome changes and phenotypic variation in budding yeast | http://www.nature.com/nature/journal/v468/n7321/extref/nature09529-s2.xls | Public. |

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
