## [Decision Letter]

Thank you for sending your work entitled “Quantitative Proteomic Analysis Reveals Posttranslational Responses to Aneuploidy in Yeast” for consideration at *eLife.* Your article has been favorably evaluated by Randy Schekman (Senior editor) and 3 reviewers, one of whom, Ivan Dikic, is a member of our Board of Reviewing Editors.

The Reviewing editor and the other reviewers discussed their comments before we reached this decision, and the Reviewing editor has assembled the following comments to help you prepare a revised submission.

Firstly we would like to thank you for the nice work that is, in the eyes of the reviewers, novel, of broad interest, and something that can be recommended. A comprehensive approach is used with two quantitative proteomic (mass spectrometry based) methods to test on a genome wide basis the idea of proteome imbalance in haploid yeast with a second copy of a chromosome (diploidy for twelve yeast chromosomes is tested!). The overall analysis revealed the existence of both transcriptionally and post-transcriptionally mediated protein expression changes indicative of slow growth as well as oxidative and metabolic stress. Due to the comprehensive nature and high technical quality of analysis (e.g. use of orthogonal growth and labeling conditions), as well as the importance of aneuploidy in a range of diseases, the manuscript will be an important resource for scientists from diverse fields ranging from yeast biology, neurobiology and cancer biology.

However, there are three main points that need to be further addressed by the authors:

1) The authors should provide more detailed insight into mechanisms of attenuation, or at least prove beyond reasonable doubt that protein degradation is the major mechanism involved. Obviously, the UBP6 deletion experiments are pointing heavily to this direction, but the two simple experiments should have been done to shed more light onto the mechanism: 1) correlation of protein attenuation with protein turnover in a simple dedicated experiment based on pulsed- or dynamic SILAC (turnover is featured highly in the Abstract, but no attempts were made to address it in the manuscript); 2) compare ubiquitylation levels of attenuated proteins in haploid, disome, and disome/UBP6- strain by performing quantitative antibody pulldowns of GG-containing peptides. This experiment would strengthen the case for protein degradation as the major mechanism of attenuation and would reveal the direct targets of UBP6 and hence provide more insight into its role in cellular response to aneuploidy.

2) The definition of the aneuploidy protein response (APR) signature is not convincing. It is based on Gene Ontology (GO) term frequencies. Many scientists use these proposed GO pathway analyses, but the term definitions are so broad and indiscriminate that the analyses are weak evidence, at best. The reviewers think that the better evidence is necessary to establish a convincing aneuploidy protein response signature. This certainly needs more than “11 of 92” altered proteins are “response to oxidative stress”.

3) In addition, in Figure 3 the reviewer has found a duplication, meaning that one picture appears twice for different data sets: The panels showing Dis XVI RNA-Seq and Dis XVI mRNA footprints data are identical. This duplication should be resolved.

---

## [Author Response]

*1) The authors should provide more detailed insight into mechanisms of attenuation, or at least prove beyond reasonable doubt that protein degradation is the major mechanism involved. Obviously*, *the UBP6 deletion experiments are pointing heavily to this direction, but the two simple experiments should have been done to shed more light onto the mechanism:*

*1.1) Correlation of protein attenuation with protein turnover in a simple dedicated experiment based on pulsed- or dynamic SILAC (turnover is featured highly in the Abstract, but no attempts were made to address it in the manuscript*.

Every experimental attempt to measure changes in the half-life of proteins encoded by the duplicated genes that are attenuated, including cycloheximide chases, ^35^S-Methionine pulse chases and pulsed SILAC, have failed. When only a fraction of a protein is unstable as is expected in the case where only unassembled subunits are degraded, half-life measurements become exceedingly difficult. The 50% of the protein that are not assembled into complexes are highly unstable whereas the other 50%, that are assembled into the complex will have very long half-lives. Thus, the fast rate of degradation of the uncomplexed pool, will be masked by the exceedingly stable pool and a bi-phasic degradation pattern very hard to see.

Comparison of the footprinting and proteomic data illustrates this difficulty. Ribosomal footprints are doubled while protein levels are not. Thus 50% of the protein must be degraded within seconds. Therefore, the total decay must consist of two rate constants, one that accounts for the long-lived stable and abundant population (k_s_), while the other describes the short-lived and undetectable population (k_u_). The total rate of degradation k is the sum of rate of degradation of stable proteins plus the rate of degradation of unstable proteins (k = k_s_ + k_u_). If k_u_ is in the range of seconds and k_s_ in minutes, then k = k_s_ + k_s_/60. Then the apparent rate of degradation is simply given by k_s_. Therefore, most experimental designs to measure changes in protein turnover will only detect the degradation of the abundant, stable and long-lived population. Consistently, experimental half-lives of most of the attenuated protein is long. To mention a few examples, attenuated genes in chromosome V (identified in Figure 4, see below) include IES5, GIM4, SCC4, BUR6 and RSM18 which show half-lives of 75, 300, 136, 300 and 66 minutes, respectively. Attenuated genes in chromosome II include ROX3, PAF1, NTC20,MED8, and HIR1 with half-lives of 123, 117, 300, 300 and 49 minutes, respectively (values obtained from (Belle et al., 2006)).

Nonetheless, to test the idea that protein degradation was responsible for the dosage compensated proteins, we examined the consequences of inhibiting protein turnover on their abundance. Protein turnover inhibition may elicit cellular responses including transcriptional feedback mechanisms that could significantly alter the proteome composition of cells. To avoid this, we inhibited the proteasome and vacuolar degradation for very short times (90 and 300 seconds) and measured protein abundances of wild-type cells and 2 disomic strains. We found that the levels of several attenuated proteins increased ranging from 4 to up to 30% upon inhibition of protein turnover even after such short times. These results provide direct evidence for protein degradation being a mechanism for the attenuation of duplicated genes. The results of these experiments have been added to the manuscript and are presented in Figure 4 and Figure 4—figure supplement 1.

*1.2) Compare ubiquitylation levels of attenuated proteins in haploid, disome, and disome/UBP6- strain by performing quantitative antibody pulldowns of GG-containing peptides. This experiment would strengthen the case for protein degradation as the major mechanism of attenuation and would reveal the direct targets of UBP6 and hence provide more insight into its role in cellular response to aneuploidy*.

Western blot analysis indicated that disomic strains do not show significant increases in ubiquitin conjugates compared to wild-type cells ([45] and unpublished results). Furthermore, because in an exponentially growing population the total amounts of attenuated proteins in disomic strains are similar to wild-type cells; it would be not expected to find significant differences in their ubiquitylation levels. Presumably, the attenuated proteins have been ubiquitinated and are rapidly degraded making them really difficult to detect and quantify. One possibility to detect changes in ubiquitination may be to combine the protein turnover inhibition plus quantitative antibody pulldowns of GG-containing peptides. Because of the results presented in Figure 4, we could expect that only 4 to 20% of total level of attenuated proteins would show changes in ubiquitination. While we realized that this could be done, these experiments are experimentally challenging will take a significant amount of time and resources to complete.

We agree with the reviewers: identifying the direct targets of Ubp6 is a critical question that could significantly contribute to our understanding of the mechanisms by which deubiquitinating enzymes regulate cellular processes. Although loss of *UBP6* significantly affects the levels of total ubiquitin conjugates in wild-type cells (17; 45) finding specific targets is a challenge and to our knowledge the physiological targets of Ubp6 have yet to be discovered. We plan to exploit the disomic strains as a tool to identify the molecular mechanisms by which *UBP6* and its loss affect cellular physiology. Because the major focus of our manuscript is on the cellular responses to aneuploidy, we think that identifying the direct targets of *UBP6* are beyond the scope of this manuscript and should be reported independently.

*2) The definition of the aneuploidy protein response (APR) signature is not convincing. It is based on Gene Ontology (GO) term frequencies. Many scientists use these proposed GO pathway analyses, but the term definitions are so broad and indiscriminate that the analyses are weak evidence, at best. The reviewers think that the better evidence is necessary to establish a convincing aneuploidy protein response signature. This certainly needs more than “11 of 92” altered proteins are “response to oxidative stress”*.

We coined the APR abbreviation to easily refer to the set of upregulated proteins specifically upregulated in the aneuploid strains. We have now changed APR for APS which changes “aneuploidy protein response” to “Aneuploidy-associate Protein Signature”. Importantly the identification of this signature provided novel insights into the cellular responses to aneuploidy. First, it lead us to discover that aneuploid strains accumulate higher levels of intracellular ROS (reactive oxygen species) compared to wild-type cells. Second, analysis of individual genes revealed that the upregulation of the APS is mediated by posttranscriptional mechanisms and that the average increase in protein levels correlates with the size of the extra chromosome, indicating that the APS is a direct consequence of the cellular imbalances resulting from the acquisition of an extra chromosome. Third, amelioration of the APS upon UBP6 loss of function led us to discover that intracellular ROS levels observed in the aneuploid cells were restored close to wild-type levels upon the deletion. Finally, we found that the APS is also present in aneuploid strains recovered from random meiosis indicating that this signature is present in all aneuploid yeast strains. Altogether, the identification of the APS which include proteins that regulate responses to cellular stress and altered metabolism revealed significant and novel insights into the cellular responses to aneuploidy.

*3) In addition, in*
Figure 3
*the reviewer has found a duplication, meaning that one picture appears twice for different data sets: The panels showing Dis XVI RNA-Seq and Dis XVI mRNA footprints data are identical. This duplication should be resolved*.

We thank the reviewers for alerting us to this mistake. We have replaced the panel with the correct one. In addition, we confirmed that the analysis presented in Figure 3 was performed with the correct data set, meaning that the mix up was restricted to the figure and that the data analysis was correct.